# Cell kinetics of auxin transport and activity in *Arabidopsis* root growth and skewing

Yangjie Hu [1], Moutasem Omary[1], Yun Hu [2], Ohad Doron[3], Lukas Hoermayer[4], Qingguo Chen[2], Or Megides[3], Ori Chekli[3], Zhaojun Ding [5], Jiří Friml [4], Yunde Zhao [2✉], Ilan Tsarfaty[3✉] & Eilon Shani [1✉]

Auxin is a key regulator of plant growth and development. Local auxin biosynthesis and intercellular transport generates regional gradients in the root that are instructive for processes such as specification of developmental zones that maintain root growth and tropic responses. Here we present a toolbox to study auxin-mediated root development that features: (i) the ability to control auxin synthesis with high spatio-temporal resolution and (ii) single-cell nucleus tracking and morphokinetic analysis infrastructure. Integration of these two features enables cutting-edge analysis of root development at single-cell resolution based on morphokinetic parameters under normal growth conditions and during cell-type-specific induction of auxin biosynthesis. We show directional auxin flow in the root and refine the contributions of key players in this process. In addition, we determine the quantitative kinetics of *Arabidopsis* root meristem skewing, which depends on local auxin gradients but does not require PIN2 and AUX1 auxin transporter activities. Beyond the mechanistic insights into root development, the tools developed here will enable biologists to study kinetics and morphology of various critical processes at the single cell-level in whole organisms.

[1] School of Plant Sciences and Food Security, Tel Aviv University, Tel Aviv, Israel. [2] Section of Cell and Developmental Biology, University of California, San Diego, San Diego, CA, USA. [3] Department of Clinical Microbiology and Immunology, Tel Aviv University, Tel Aviv, Israel. [4] Institute of Science and Technology Austria, Klosterneuburg, Austria. [5] The Key Laboratory of Plant Development and Environmental Adaptation Biology, Shandong University, Qingdao, Shandong, China. ✉email: yundezhao@ucsd.edu; ilants@tauex.tau.ac.il; eilonsh@tauex.tau.ac.il

Plants are sessile multicellular organisms that depend on finely tuned distribution of small signaling molecules to coordinate cellular responses that regulate their growth and development. Signaling regulated by the plant auxin hormone indole-3-acetic acid (IAA) governs nearly all processes in a plant's life[1–3]. Amongst others, regulation of primary root growth, lateral and adventitious root formation, vasculature patterning, and root hair elongation are all dependent on auxin[4,5]. Auxin regulates cell division, elongation, and differentiation[3], and it is involved in root responses to the environment such as gravitropism[6], hydro-patterning[7], and xerotropism[8]. The developmental and physiological responses are a result of tightly regulated auxin gradients within plant tissues[9,10]. The auxin gradients are primarily established by local auxin biosynthesis and the combined activities of auxin influx and efflux carrier proteins[11]. Although it is not entirely clear which cell types produce IAA, recent studies have shown that the tryptophan aminotransferase TAA1 (also known as WEI8), TAR2 enzymes, which convert tryptophan to indole-3-pyruvic acid, and YUCCA enzymes, which convert indole-3-pyruvic acid to bioactive IAA[12–15] are required for tightly controlled auxin levels and root meristem maintenance[16,17]. Current models of auxin transport in the root suggest that auxin flows down the root apex through the stele cells depending on PIN1 and PIN7 auxin efflux carriers[18,19] and members of the AUX/LAX family of auxin importers[20]. Auxin is refluxed back by basipetal transport through the epidermis in a PIN2- and AUX1-dependent manner[6,21,22]. These fluxes, often referred to as a reversed fountain, are essential for specifying the position of the quiescent center (QC) and developmental zones[9,18,23]. The current model is largely based on the expression patterns and subcellular localizations of PIN and AUX/LAX transporters[24–26], and there is limited direct evidence of auxin flow and auxin activity at the cellular level. Here, we used an inducible multi-cell-type-specific auxin biosynthesis system and custom image-analysis tools to quantitatively characterize auxin movement and activity in high spatial and temporal resolution, shedding light on auxin-dependent kinetic parameters of root growth and skewing.

## Results

### Auxin production in certain cell types significantly affects root growth.

To understand how localized auxin production affects auxin movement and activity, we generated constructs that express YUC1 and TAA1 under the control of several different cell-type-specific inducible (estradiol) promoters and introduced them into plants expressing the auxin reporter DR5:VENUS[27,28] (Fig. 1a, b). Simultaneous expression of YUC1 and TAA1 results in IAA synthesis from tryptophan[12,14,15]. Expression of GFP driven by each promoter showed the expected expression patterns: pWER – epidermis, pSCR – endodermis, pSHR – stele, pAPL – phloem (protophloem, companion cells, and metaphloem sieve elements[29]), and pWOX5 - QC[10,30] (Fig. 1d). Importantly, no significant differences in root length compared to control plants were observed for any of the lines in the absence of estradiol treatment at day 4 and day 11, with the exception of the SHR promoter, which showed a mild effect at day 11 (Supplementary Fig. 1). These data demonstrate that the system is not leaky. The ectopic overproduction of auxin in the epidermis, endodermis, and stele all resulted in very short roots in comparison to non-treated control plants, whereas production of auxin in phloem and QC caused relatively mild responses (Fig. 1c, e). The relatively weak response to auxin production in the phloem may result from the slight YUC1-TAA1 transcriptional activation (4.5-fold change following 1.5-h estradiol treatment). Strong YUC1-TAA1 transcriptional activation (ranging from 30- to 104-fold) was observed in the other four tissues

(Supplementary Fig. 2). In order to test whether TAA1 substrate is the rate-limiting factor in the process, we applied L-tryptophan to different cell-type-specific IAA synthesis lines and tested their root growth response. In general, applying different concentrations of tryptophan (0, 10, 50, or 100 µM) did not affect the auxin-mediated root growth inhibition. Only the highest concentration of tryptophan (100 µM) caused a slight inhibition in root growth of pAPL and pSCR lines (Supplementary Fig. 3). Altogether, the results indicate that ectopic production of auxin in distinct tissues leads to significant growth inhibition over a timescale of days.

To test whether auxin activity is permutated away from the site of production, we induced synthesis of auxin in specific cell types and monitored DR5:VENUS activity over time and space. The synthetic DR5 reporter indicates an auxin response. As expected, we observed auxin responses in tissues where auxin was produced. In addition, lines expressing YUC1-TAA1 in the epidermis, QC, stele, and endodermis (6- or 24-hour estradiol treatment) showed significant and relatively similar DR5:VENUS patterns in the epidermal cell layer and differentiation zone compared to the control plants (Fig. 1f, g). The strong activation of DR5:VENUS in the elongation/differentiation transition zone following IAA production in the QC suggests that auxin movement is rapid. Importantly, no changes in any of the promoter-specific expression patterns were detected following production of IAA (Supplementary Fig. 4), suggesting that IAA synthesis did not spatiotemporally change during the 6 h of the experiment. Together, these results imply that auxin rapidly moves through the root in a well-coordinated and robust manner.

### Single-cell nucleus tracking approach reveals morphokinetics of Arabidopsis root growth and tip skewing.

It is likely that much of the response we observed when auxin production was induced in one cell type actually results from an accumulated response following auxin transport, and there is a need for a high-resolution technology to evaluate the rapid growth response induced by the hormone. Several techniques have been applied in recent years to track root growth such as light imaging, confocal and light sheet microscopies, MRI, and luminescence[31–35]. Here we developed a single-cell 4D imaging (X, Y, Z over time) method and image analysis pipeline to monitor nuclear morphokinetic growth responses. First, nuclei were labeled with the 35S:H2B-RFP fluorescent marker[36], and then, using confocal microscopy, we monitored root growth with high spatiotemporal resolution by collecting images at 40 time points over 6 h (every 8.75 min). Data were collected in two channels: red for nuclei and yellow for DR5:VENUS. To create 3D images, we used Z-stacks of 30 slices and four tiles to image all three developmental zones. For each 4D movie, we detected around 1000 nuclei from the meristem, elongation, and differentiation zones (Fig. 2a, Supplementary Movies 1–3). Using a custom MATLAB code, we generated concatenated data representing independent roots and thousands of cells.

Monitoring cell velocity in three dimensions over time showed that nuclei in the elongation zone steadily move at a velocity of approximately 50 µm/h in the Y dimension. Based on our 4D imaging and analysis infrastructure, slight changes in root velocity could be quantified, and we demonstrated that the nuclei move approximately 10 µm/h faster in the X and Z planes in the root meristem zone than in the elongation zone (Fig. 2b). The increase in cell velocity at the root meristem zone in X and Z planes likely represents root skewing. Roots wave and skew when grown on a slanted impenetrable media[37]. Skewing occurs when the steady-state growth direction of a root deviates from the direction of the gravity vector. This is believed to result from endogenous signaling

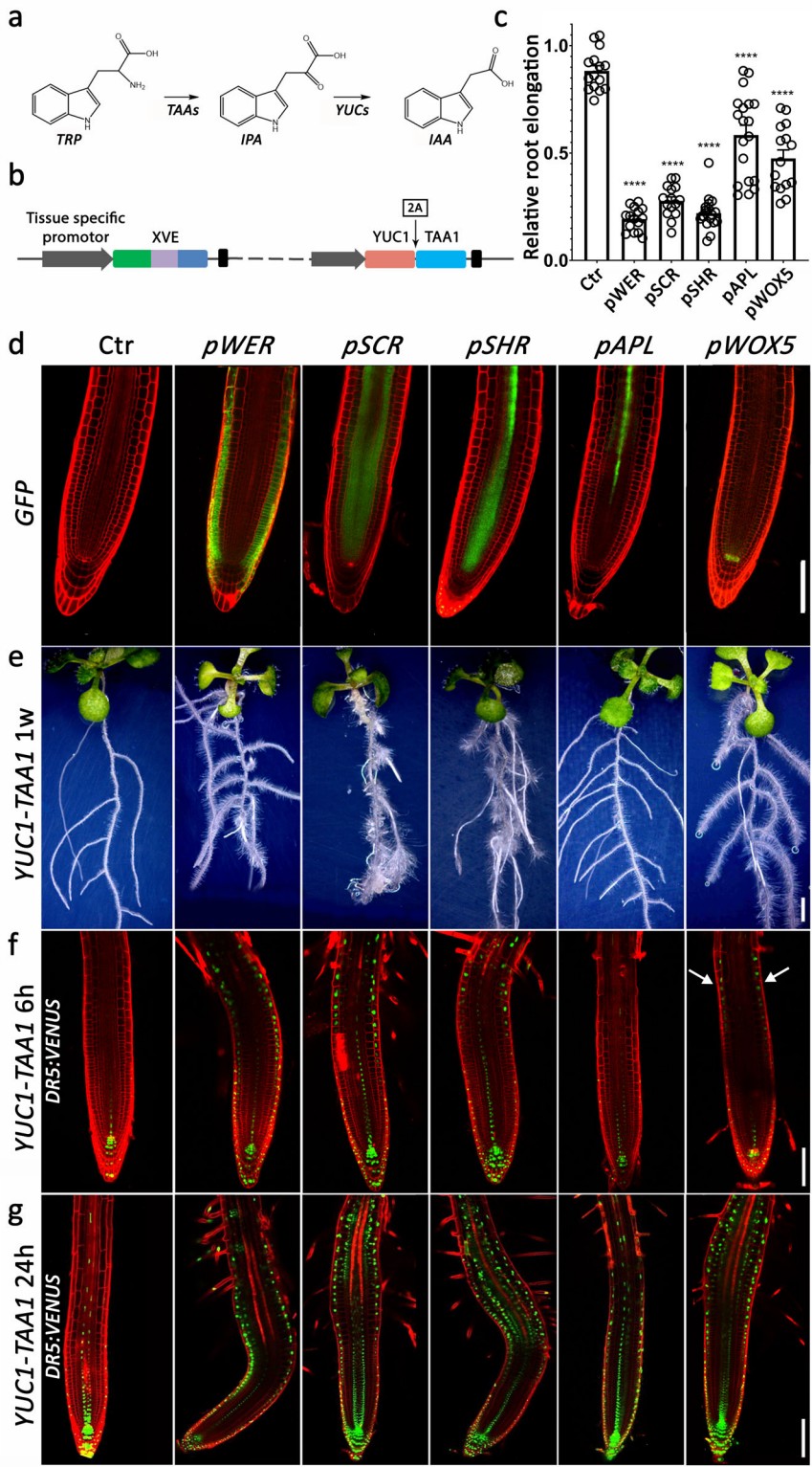

originating in external cues such as gravity and touch[38,39]. Supporting this hypothesis, the sum of velocity in all three planes (velocity total) is comparable to the velocity in the Y dimension in the elongation zone (approximately 60 μm/h) (Fig. 2b). Similarly, analyses of kinetics of X and Z dimensions for nuclei acceleration revealed significant acceleration of 5–100 μm/h$^2$ in the root meristem (Supplementary Fig. 5).

We next evaluated whether cells move in a coordinated fashion over time. For every time point, we marked all the cells within a 200-μm radius from a chosen center cell. For every cell in the perimeter, we calculated the cosines of the velocity angles in relationship to the chosen center cell. In the differentiation zone, there was little coordination. In contrast, in the elongation zone, nuclei showed a 6-fold increase in coordinated motility (Fig. 2c). Cells in the meristematic zone showed no coordination. We hypothesize that the low coordinated motility in the meristem might reflect stochastic cell division or local root skewing of cells with opposing vectorial motion positioned on opposite sides of

**Fig. 1 Cell-type-specific expression of YUC1 and TAA1 impedes root growth and reveals rapid auxin movement. a** TAAs- and YUCs-Trp-dependent IAA biosynthesis pathway. **b** Schematic diagram of the cell-type-specific promoter XVE vector. Upon estradiol treatment, the XVE fusion protein transcriptionally activates expression of YUC1 and TAA1 in specific cell types. The sequence encoding the 2 A self-cleaving peptide sequence is located between YUC1 and TAA1. **c** Root elongation of seedlings expressing cell-type-specific YUC1-TAA1 (YUC1-2A-TAA1) vectors. Seeds were sown on MS, and seedlings were transferred on day 4 to plates containing 5 μM estradiol. Root length was scored after 7 days and normalized to root length of control plants (no estradiol treatment) at the same age (Supplementary Fig. 1). Shown are means (±SE), $n \geq 15$ plants (Ctr, pWER, pSCR, and pWOX5 = 15, pSHR = 18, pAPL = 19), ****$P < 0.0001$ by two-tailed $t$ test. **d** Expression patterns of GFP produced from cell-type-specific promoters in 5 day-old seedlings at 24 h after estradiol induction (5 μM). pWOX5 is pWOX5:mCHERRY. GFP or mCHERRY are not part of the YUC1-2A-TAA1 construct. Scale bar = 100 μm. Propidium iodide staining is shown in red. **e** Representative images of seedlings that express cell-type-specific YUC1-TAA1 at 1 week (1 w) after estradiol treatment (5 μM). Only the upper part of the root is shown. The roots are not shown as a whole. Scale bar = 2 mm. **f–g** DR5:VENUS response following cell-type-specific auxin production. Plants were treated on day 5 with estradiol (5 μM) for **f** 6 h and **g** 24 h. White arrows indicate DR5:VENUS activation in epidermal-elongating cells of pWOX5:XVE:YUC1-TAA1. Green: DR5:VENUS signal; red: propidium iodide dye marking cell walls. Scale bar = 100 μm. The experiments were independently repeated three times.

the root meristem diameter. In addition to the shifts in root meristem zone X- and Z-plane velocity (Fig. 2b), data from three independent experiments suggest that the low coordinated motility in the meristem is driven by root skewing. First, long-term vertical-stage microscopy tracking (19 and 25 h) showed meristem-zone skewing dynamics (Supplementary Fig. 6, Supplementary Movies 4 and 5). Second, nuclei tracking showed clear opposing vectorial motion over time (Fig. 2d, Supplementary Movie 6), independent from media rigidity (Supplementary Fig. 7). Third, we tested the frequency of cell divisions in the meristematic zone in a 20-hour vertical-stage microscope experiment, comparing mock and NAA treated samples. Whereas NAA treatment caused only a mild inhibition in cell division under these settings (Supplementary Fig. 8), root skewing was completely repressed (Supplementary Movies 4 and 5). Together, these data suggest that the low coordinated motility in the meristem is likely driven by root skewing. Further, the results show that the nuclei tracking approach can quantitatively report on multiple novel kinetic parameters in 4D at the single-nuclei resolution.

**Cell-type-specific auxin production differentially affects root kinetics and skewing as shown by single-cell image analysis.** To understand how cell-type-specific auxin production affects single-cell kinetics, we introduced five inducible tissue-specific promoters WER, SCR, SHR, APL, and WOX5 that drive YUC1 and TAA1 expression into the 35S:H2B-RFP; DR5:VENUS background (homozygous for all three constructs). As analysis of the QC-specific line (pWOX5:XVE:YUC1-TAA1) suggested rapid shootward movement of auxin, we generated the pCLE40:XVE:YUC1-TAA1; DR5:VENUS line, which restricts IAA production to the columella cells and differentiated stele[40]. Expression of YUC1-TAA1 under control of the pCLE40 promoter resulted in comparable activation of YUC1 expression as observed when the gene was under the control of the WOX5 promoter (~26 fold-change) (Supplementary Fig. 2). Three independent roots of six inducible tissue-specific YUC1-TAA1; DR5:VENUS; 35S:H2B-RFP plants and three control plants were imaged for 6 h following estradiol induction. Using the cell morphokinetic infrastructure, kinetic and morphology parameters were analyzed. Velocity, acceleration, displacement-squared, instantaneous angle, and coordinated motility were significantly different between plants with tissue-specific induction of IAA biosynthesis compared to the control; whereas eccentricity and nuclei area were less responsive (Fig. 3a).

Our temporal analysis of the average kinetic parameters showed that root cells increase in velocity during the first 30 min after induction, followed by a steep decline, suggesting that initial relatively low IAA concentration enhances root growth and

then the gradual increase in auxin levels and distribution consequently inhibit the root cell velocities. Although in the pCLE40 line, the expression of YUC1 and TAA1 was restricted to the columella cells and stele, the roots showed a rapid response similar to lines in which YUC1 and TAA1 were expressed in stele (pSHR) and epidermis (pWER) (Fig. 3b, c). Cluster analysis for multiple kinetic and morphological parameters (Supplementary Table 1) revealed that plants with induced IAA biosynthesis in the phloem (pAPL) cluster together with the control, distinct from all other lines. Surprisingly, despite distinct expression patterns, data on pCLE40 (columella cells and stele) and pWER (epidermis) lines clustered in one group, with distinct different morphokinetic characteristics from the control data. Data on pSCR (endodermis) and pWOX5 (QC) lines were clustered in a second group and pSHR (stele) data were between these two clusters (Fig. 3c, Supplementary Fig. 9). The data are in-line with the root concentration-dependent response to exogenous IAA treatment (Supplementary Fig. 10). The rapid but transient inhibition in the velocity of cells in response to 20 nM IAA treatment demonstrates the spatiotemporal sensitivity of the monitoring system and suggests that pAPL non-responsiveness is likely not because of insufficient activation of IAA. Since the data represent kinetic and morphological responses in the first 6 h following IAA production in given cell types, differences likely reflect auxin synthesis levels and sites of auxin action as well as rapid auxin transport. Although the auxin response in the epidermis is in-line with previous observations[41,42], the overlapping activity in epidermis (pWER), columella cells and stele (pCLE40), and QC (pWOX5) suggests that there is rapid auxin movement from root tip tissues (columella and QC) to the elongation zone, which is 300–400 μm away.

Nuclei in the pWOX5 seedlings showed similar velocity in terms of time and space to nuclei in the pSHR line (stele); pCLE40 (columella cells and stele) and pWER (epidermis) lines had steeper responses (Fig. 4). The DR5:VENUS intensity, starting from the elongation zone and expanding both directions over time was not entirely correlated with the kinetics response. Whereas changes in velocity and acceleration were detected within minutes of estradiol treatment (Fig. 4, Supplementary Fig. 11), the DR5:VENUS signal was observed only after about 2.5 h, consistent with previous publications[43]. Similarly, whereas the pSHR line with IAA induction in the stele showed a stronger DR5:VENUS intensity than pCLE40 and pWER lines with expression in columella cells and stele and in epidermis, respectively, decay in velocity was slower (Fig. 4). Interestingly, coordinated motility was dramatically increased in the root meristem zone when auxin biosynthesis was upregulated in all cell types except phloem (pAPL). The highly coordinated motility in the meristematic zone likely reflects the inhibition of root skewing, suggesting that auxin is a key player in this process. This is supported by the rapid augmentation in meristem zone coordinated

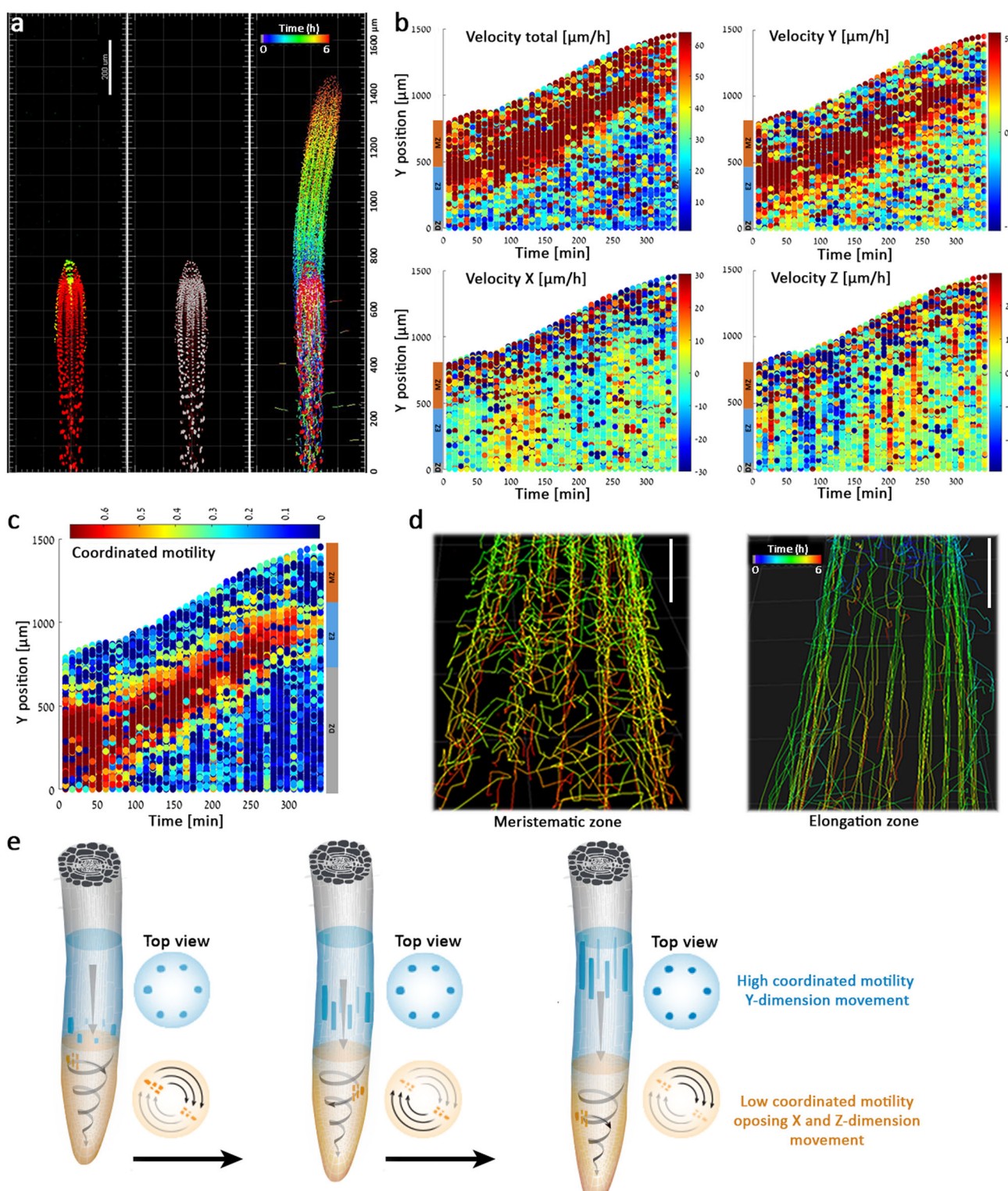

motility in response to low- and high-concentration IAA treatments (Supplementary Fig. 10a). Root tracking showed that whereas the control and the *pAPL* lines had high skewing motions, *pSCR*, *pWOX5*, *pCLE40*, *pWER*, and *pSHR* lines had motion limited to the Y axis (Supplementary Fig. 12). Moreover, long-term vertical-stage microscopy tracking (25 h) showed that auxin treatment or cell-type-specific auxin induction completely inhibited root skewing (Supplementary Fig. 6, Supplementary Movies 4, 5, and 7). These results suggest that root skewing is tightly linked to root growth in an auxin-gradient-dependent manner.

**Mapping IAA movement in the root.** Previous studies demonstrated that polar subcellular localization and tissue-specific expression patterns of PINs and AUX/LAX proteins are associated with auxin movement[18,44,45]. In order to map auxin movement in high resolution, we monitored the *DR5:VENUS* response (indicative of auxin response sites) following local IAA production in mock-treated control seedlings and in seedlings treated with the polar auxin transporter inhibitor NPA, which is considered as an inhibitor of PIN proteins and might also target TWD1 (TWISTED DWARF 1) and ABCB transporters[46,47], or

**Fig. 2 Single-cell tracking approach quantitatively reports on root growth and skewing. a** Left: *pWOX5:XVE:YUC1-TAA1; DR5:VENUS* roots without estradiol treatment (control) were imaged in 3D over time. Red: *35S:H2B-RFP* (marking nuclei); green: *DR5:VENUS*. Middle: Single-nuclei image analysis detection; each nucleus is indicated by a gray dot. Right: Tracking of individual nuclei over time and space. Time is indicated by the rainbow scale (0–6 h). The experiment was independently repeated three times. Scale bar = 200 µm. **b** Velocity maps over time. Root tip is positioned upwards. On the color scale on the right, high velocity is red; low velocity is blue. The color scale on the left indicates developmental stages (MZ meristem zone, EZ elongation zone, DZ differentiation zone). **c** Coordinated motility over time. On the color scale on the top, red indicates strong coordination; blue indicates no coordination. The color scale on the right indicates developmental stages (MZ meristem zone, EZ elongation zone, DZ differentiation zone). **d** Single-nuclei tracking routes of cells positioned at the meristem (left) and elongation (right) zones. Meristematic zone tracking shows opposing vectorial motion of cells positioned in opposite sides of the root diameter. $T_O$ is 20 min following estradiol treatment; rainbow scale indicates time (0–6 h). The experiment and analysis were independently repeated three times. Scale bars = 50 µm. **e** Schematic representation of root skewing and coordinated motility in the elongation and meristem zones. In the elongation zone (blue), cells move primarily in the Y-dimension, therefore showing high coordinated motility. The skewing of the meristem zone (orange) generates motion in X, Y, and Z-dimensions. Cells positioned on opposing sides of the root meristem move in opposite directions, therefore showing low coordinated motility.

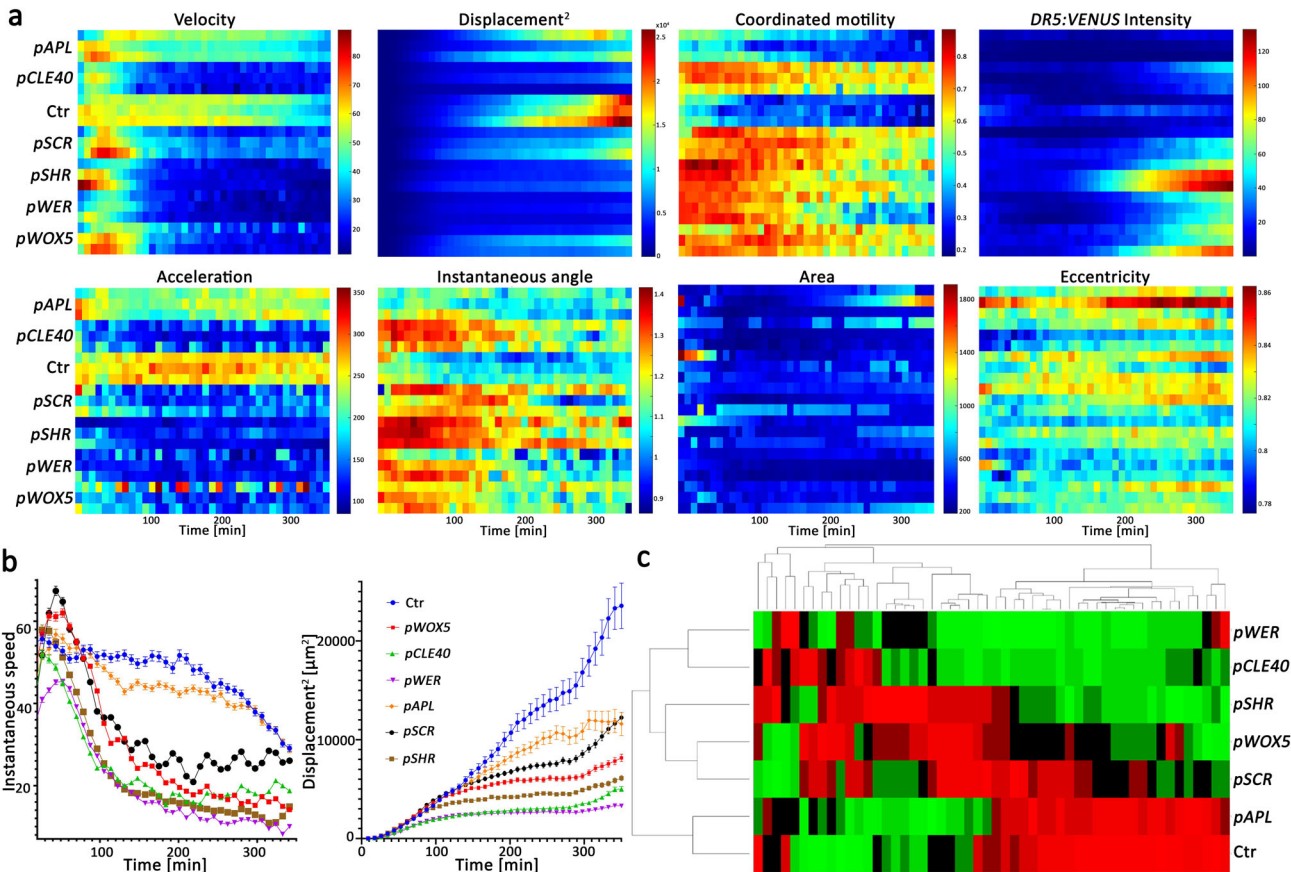

**Fig. 3 Root kinetics in response to cell-type-specific IAA induction. a** Heat maps of 8 root kinetic and morphologic parameters in response to cell-type-specific auxin production. Red indicates high and blue indicates low values. The data are average values for the different parameters collected from all root nuclei over time including meristem, elongation, and differentiation zones. Data were concatenated from three independent root movies for each line, each with ~1000 nuclei. **b** Instantaneous speed and cell displacement over time for the indicated lines. Shown are means (±SD), $n \geq 300$. **c** Hierarchical clustergram analysis of the indicated lines grouped by morphokinetic parameters. Ctr indicates control mock-treated *pWOX5:XVE:YUC1-TAA1; DR5:VENUS* seedlings. For complete list of all parameters see Supplementary Table 1 and Supplementary Fig. 9. Data were concatenated from three independent root movies for each line, each with ~1000 nuclei. The $n$ number for each line and time point is indicated at Source data file. $T_O$ is 20 min following 5 µM estradiol treatment.

with 1-NOA, which is considered as an AUX1/LAX inhibitor[24,48]. In seedlings in which IAA biosynthesis was induced in QC (*pWOX5*), NPA and 1-NOA strongly blocked shootward auxin movement (Fig. 5a–c, black arrow), confirming previous studies showing that shootward directed auxin movement is dependent on PINs and AUX/LAX proteins[45,49]. Single-cell image analysis further demonstrated how 1-NOA (to a large extent) and NPA (to a lower but significant extent) delayed root growth inhibition generated by QC-specific auxin induction. Interestingly, 1-NOA

partially repressed *pWOX5:XVE:YUC1-TAA1*-dependent root skewing inhibition (Supplementary Fig. 13).

Interesting results were obtained following induction of IAA in the stele (*pSHR*). Whereas NPA inhibited the *DR5:VENUS* response in epidermis and QC to a large extent, 1-NOA only prevented the *DR5:VENUS* response in the epidermis but not in the QC (Fig. 5a–c, red arrows), implying that rootward, stele-mediated, auxin flow is a PIN-regulated process that does not require the activity of the AUX/LAX proteins. Importantly,

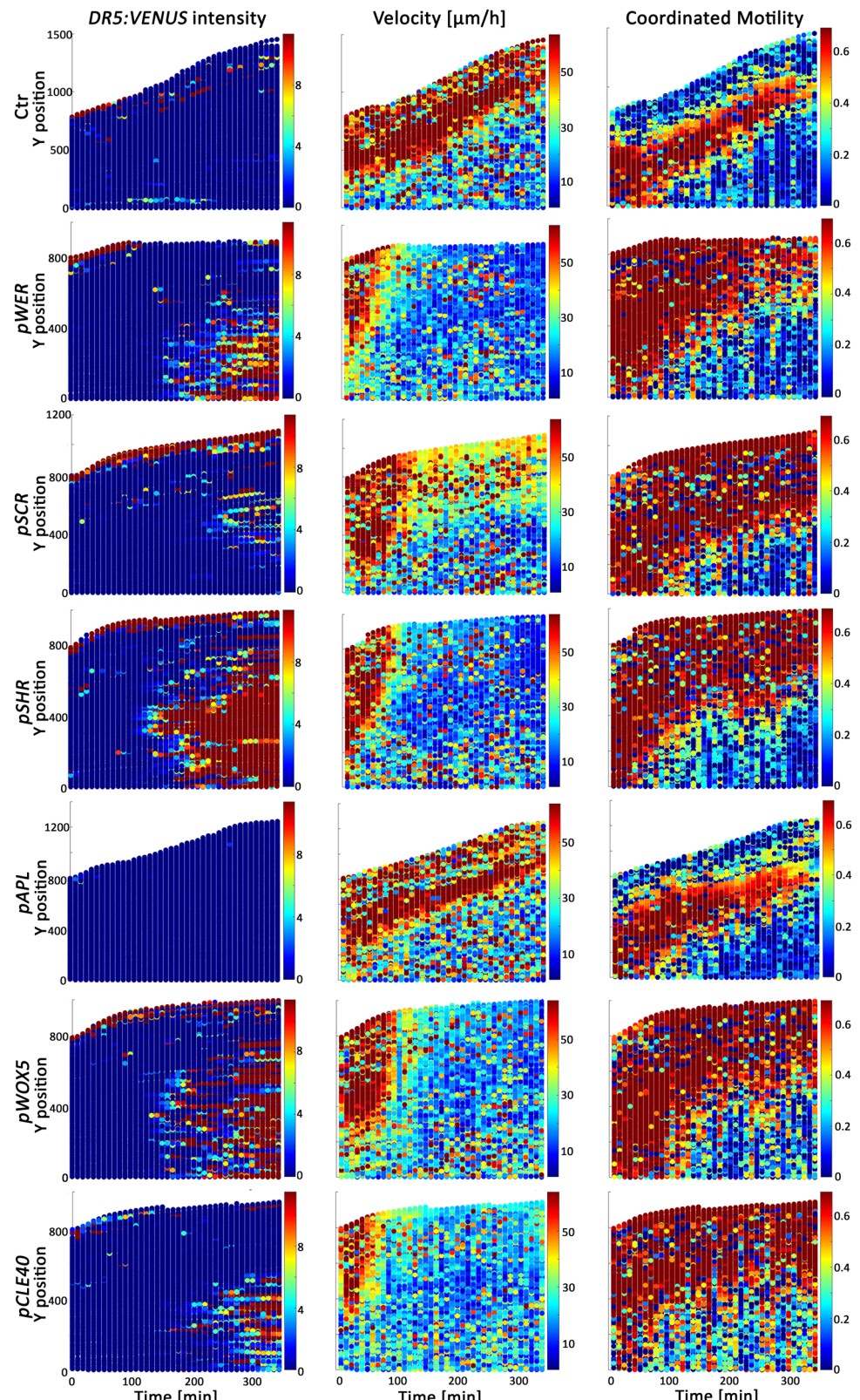

**Fig. 4 Cell-type-specific auxin production kinetic response over time and space in the root.** Single-cell mean *DR5:VENUS* intensity (left), velocity (middle), and coordinated motility (right) maps following cell-type-specific auxin induction. Ctr indicates control plants, which are mock-treated *pWOX5: XVE:YUC1-TAA1; DR5:VENUS* seedlings. Root tip faces upwards. Data are a concatenate of three independent movies of roots of each genotype, each with ~1000 nuclei. $T_0$ indicates 20 min after 5 μM estradiol treatment.

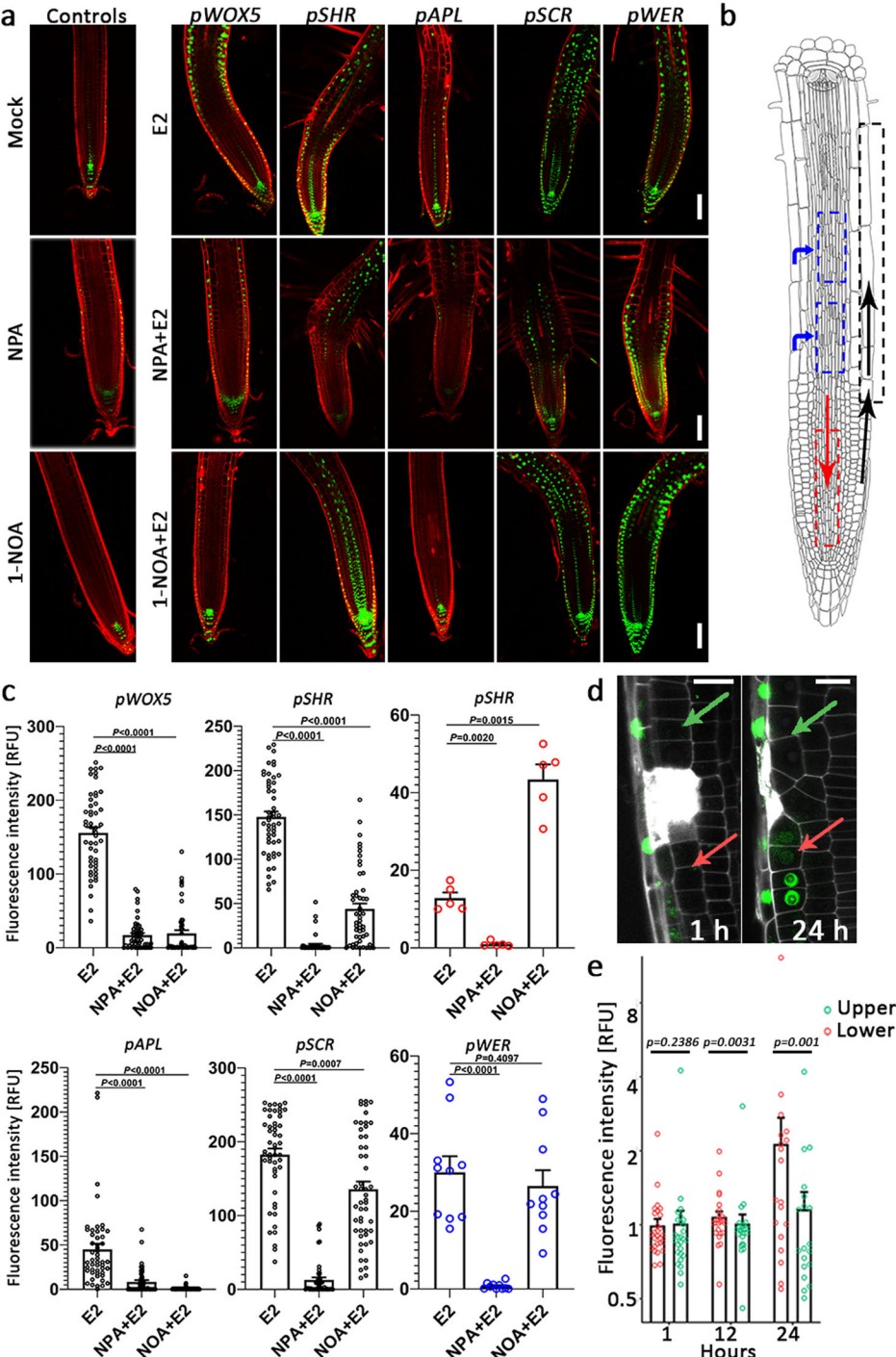

**Fig. 5 NPA impedes auxin flux rootward, shootward, and inward, whereas 1-NOA affects only shootward auxin movement. a** Cell-type-specific IAA inducible lines were grown for 4 days on MS agar before transfer to agar containing vehicle (Mock) or 25 µM NPA or 50 µM 1-NOA for 20 h. Seedlings were then mock treated or treated with 5 µM estradiol (E2) for 24 h and imaged. Representative root confocal microscope images are displayed. Propidium iodide staining is shown in red, *DR5:VENUS* activity in green. Scale bar = 100 µm. **b** Schematic representation of root geometry. The dashed color-coded boxes define regions of interest for *DR5:VENUS* signal quantification. The arrows represent auxin fluxes. **c** Fluorescence intensity quantification of *DR5: VENUS* roots treated as described in panel **a**, measured at regions of interest corresponding to colors shown in panel **b**. Shown are means (±SE), $n = 5$ plants, $P$ value two tailed $t$ test is indicated for each analysis. **d** *DR5:VENUS* expression (green) in cells above and below laser-assisted elimination of epidermis, 1 and 24 h after ablation. Cell walls and ablated cells are stained with propidium iodide (10 µM) shown in white. Scale bar = 15 µm. **e** Fluorescence intensity quantification of *DR5:VENUS* below and above ablations as described in panel **d**. $n = 19$–27 cells from 3 independent experiments for each category ($n = 27$ for 1 h, $n = 25$ for 12 h and $n = 19$ for 24 h treatments). Shown are means (±SE). Statistical significance between upper (green arrow) and lower (red arrow) cells was computed from a paired, two-tailed Wilcoxon test, considering the non-normal distribution of the data.

inducing IAA biosynthesis in the epidermis (*pWER*) resulted in *DR5:VENUS* induction in the stele differentiation zone, a process that was inhibited by NPA but not 1-NOA (Fig. 5a–c, blue arrow), suggesting that IAA flux into the stele is regulated by PINs but not by AUX/LAX proteins.

Next, we wanted to test whether blocking of auxin movement in specific cells would generate an auxin maximum prior to the jam in a position that depends on auxin flow. Laser-assisted elimination of epidermis cells showed *DR5:VENUS* induction below the wound. The induction was detected 12 and 24 h following laser cell ablation, consistent with auxin moving from the root meristem shootwards through the epidermis (Fig. 5d, e). Laser-assisted elimination of stele cells did not show *DR5:VENUS* induction at similar time points (Supplementary Fig. 14). The non-significant *DR5:VENUS* induction above the wound may be partially masked by limited wounding not covering the entire auxin stream in the stele and indirect induced local auxin production in the root meristem within hours[50].

**PIN2 and AUX1 are required for directional auxin movement and root growth but not root skewing.** We next utilized the cell-type-specific system for induction of auxin synthesis combined with the single-cell image analysis approach to test the requirement of known auxin transporters in root growth and tip skewing. We generated lines expressing the specific promoters in the background of the *pin2* and *aux1* knockouts that express the R2D2 auxin reporter. R2D2 was chosen as it ratiometrically reflects auxin concentration much more rapidly than *DR5:VENUS*[43,51]. As expected, auxin production in the epidermis (*pWER*) resulted in strong degradation of the DII-VENUS protein and inhibited root instantaneous speed in similar manners for *aux1* and *pin2* and respective controls (Fig. 6). These results suggest that auxin produced in the epidermis layer acts locally, at sites where auxin is required for root elongation, and does not require the activity of PIN2 or AUX1. In contrast, induction of auxin production in the stele (*pSHR*) or QC (*pWOX5*) in the background of the knockouts resulted delayed decay of R2D2 fluorescent intensity and delayed root growth kinetics (Fig. 6a–f, Supplementary Fig. 15). The results are in line with the NPA and 1-NOA *pWOX5:XVE:YUC1-TAA1*; *DR5:VENUS* results (Fig. 5), and together indicate that PIN2 and AUX1 are both required for auxin movement through the epidermal cells to maintain control levels of root growth velocity. Since our results indicate that root skewing is regulated by auxin gradients, we tested whether PIN2 or AUX1 is required for this process. *pin2* and *aux1* mutant seedlings have skewing angles similar to that of the control (Fig. 6g, h, Supplementary Fig. 16), and single-cell X and Z velocity scatter plots showed no difference between *pin2* and *aux1* mutant seedlings compared to controls. However, velocity in X and Z dimensions showed significant reductions when auxin synthesis was induced in the epidermis and a mild response when induced in the stele (Fig. 6i, j, Supplementary Fig. 17). This suggests that PIN2 and AUX1 are not involved in root skewing and that additional regulators of auxin distribution involved in root skewing are yet to be found.

## Discussion

In 1881, Charles Darwin laid the conceptual groundwork for understanding plant movements such as circumnutation[52]. Root circumnutation and skewing due to endogenous cues, gravity, and contact with the medium result in a spiral growth pattern[53]. Space flight research suggests that the force of gravity is not needed for waving and skewing patterns of *Arabidopsis* roots grown on solid surfaces[54]. These movements are presumably adaptive because they facilitate discovery of the easiest course

through the soil[38]. Although the molecular factors that regulate root skewing have not been identified, it was recently shown that impairment in sensing of karrikin, but not in sensing of strigolactone, enhances root skewing in *Arabidopsis*[55]. The results we report here show that auxin is a key player in root meristem skewing. It is thought that different cues perceived by the root meristem as it grows translate into a change in auxin flux within the root cells by transfer of auxin to the elongation zone[20,49]. Whereas knockouts of *aux1* and *pin2* were suggested to have impeded root waving[37], our results show that AUX1 and PIN2 are not required for root skewing. The results raise the possibility that root waving and root skewing are molecularly disconnected. It will be interesting to test whether karrikins are dynamically distributed in a polar fashion in the root or rather work upstream or downstream of auxin in root skewing regulation, and it will be important to identify the molecular factors regulating auxin redistribution during root skewing.

*YUC1-TAA1* driven by promoter *APL* (phloem) showed the weakest response in roots. The construct is active since the seedlings show strong enhancement in hypocotyl elongation (not shown). Several factors may have caused the weak root growth-inhibition response. First, the inducible promoter *APL1* (*pAPL:XVE:YUC1-TAA1*) generated a relatively weak *YUC1* induction (4.5-fold change following 1.5-h estradiol treatment) compared to 30- to 104-fold observed using the other four promoters (Supplementary Fig. 2). Thus, a combination of a relatively small number of cells expressing *APL* and low expression may cause the low levels of activation in the root. Since roots treated with 20 nM IAA showed significant alterations in both root growth and skewing, we speculate that non-responsiveness of *pAPL:XVE:YUC1-TAA1* is due to factors in addition to low auxin induction. Additional explanations may be low penetration of estradiol into the phloem (protophloem, companion cells, and metaphloem sieve elements[29]), rapid movement/transport of IAA out of the phloem cells, rapid metabolism (degradation and conjugation) of IAA in phloem cell, low perception or signaling in phloem cells, relatively low levels of tryptophan (TAA1 substrate), or the combined effects of several of these processes. Our data presented is in agreement with a recent report that demonstrated that auxin flux homeostasis through the phloem is well buffered[56].

Our use of the cell-type-specific induction systems of IAA synthesis, combined with single-cell image analysis revealed auxin motion from the IAA point of view. In the past, multiple studies have convincingly utilized NPA and 1-NOA demonstrate requirements for PIN, AUX/LAX, and ABCBs in directional auxin transport[24–26]. Our analysis, using cell-type auxin induction, showed auxin movement in three directions: rootward through the stele, shootward through the epidermis, and back into the stele at the differentiation zone. We demonstrated that all three directional auxin flow patterns (reflected by *DR5:VENUS* and *R2D2* response) are NPA and PIN dependent. Only the shootward auxin movement was 1-NOA or AUX1 dependent. It is important to note that these results should be carefully interpreted as the biochemical specificity of 1-NOA or AUX1 are not completely understood, and the results may reflect effects secondary to developmental effects of these inhibitors.

The LAX2 auxin transporter has been implicated in the auxin rootward transport. The results presented here suggest that the AUX/LAX transporters do not play a major role in rootward auxin movement. The difference in observations may result from non-specific and inefficient activity of 1-NOA toward LAX2. Additionally, the production of IAA using the *SHR* promoter (driving inducible *YUC1-TAA1*) may lead to non-physiological levels of IAA that float the AUX/LAX system. Finally, differences may result from the experimental setups. Whereas previous observations and modeling largely relied on analyses of *LAX2* expression

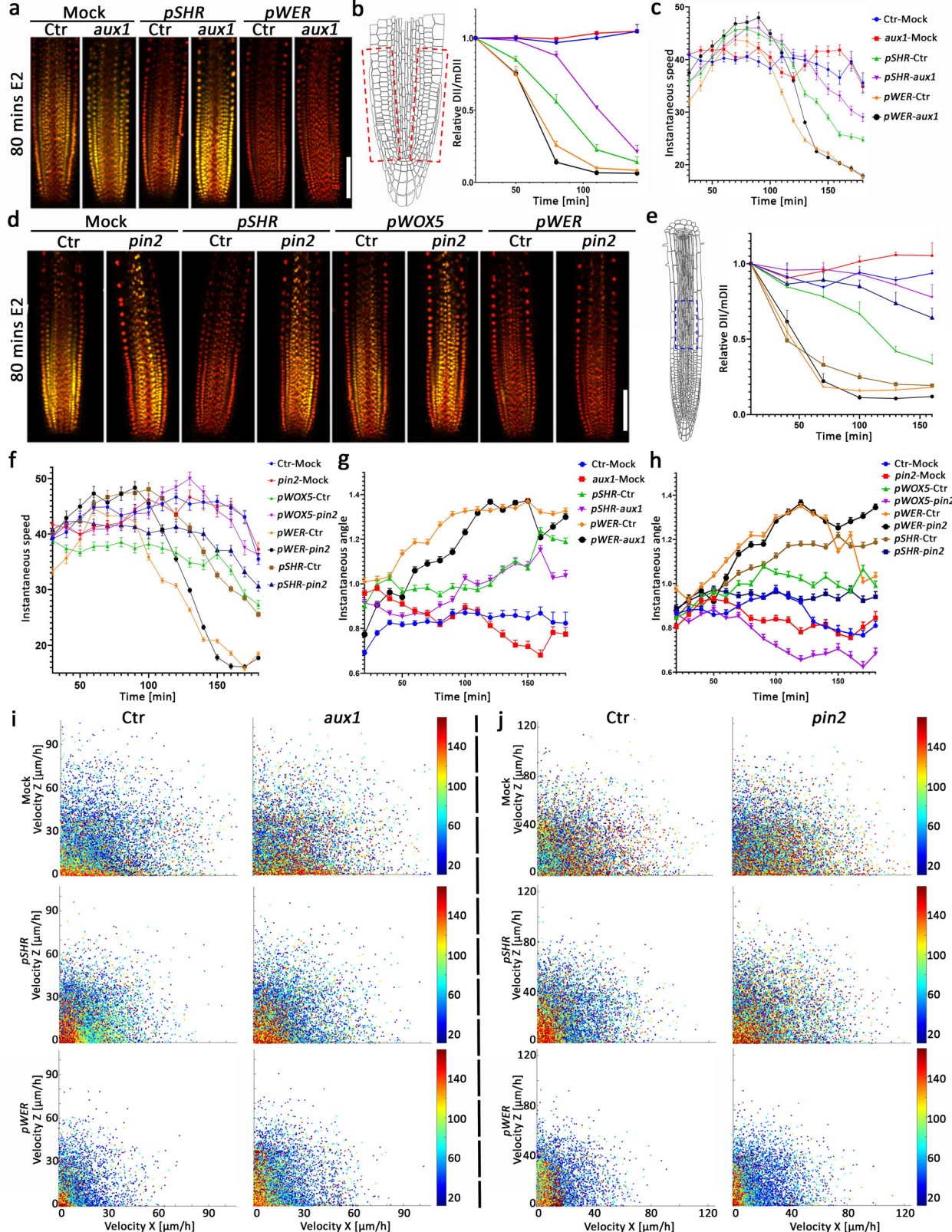

patterns[20,26], in the work presented here, we induced auxin biosynthesis in one location and tracked its response and activity over time. The combination of cell-type-specific auxin induction and transporter mutants allowed us to test activity with spatial auxin transport resolution. The results confirmed requirements for PIN2 and AUX1 in shootwards auxin flux and highlighted the

elongation epidermis cells as the main go-to and response sites. Importantly, the PINs, or possibly ABCBs, necessary for the auxin flux back into the stele remain unidentified.

The quantitative morphokinetic parameters presented here could be incorporated into mathematical models[20] to define auxin activity on root growth, response to the environment, and

**Fig. 6 AUX1 and PIN2 are required for directional auxin transport into the epidermis but not for auxin-mediated root meristem skewing. a** DII ratiometric signal in the indicated *YUC-TAA1 R2D2* lines in the *aux1* background. Presented are merged confocal images of mDII (red) and DII (yellow). Scale bar = 100 μm. **b** Quantification of relative DII/mDII following 5 μM estradiol and mock treatments of indicated *aux1*-mutant lines. The red boxes indicate regions used to quantify the DII/mDII ratio. Shown are means (±SE) of two independent roots, each with two regions of interest (*n* = 4). **c** Quantification of root nuclei instantaneous speed following 5 μM estradiol treatment of the indicated *aux1*-mutant lines. Data points are concatenated means (±SD) of ~1000 nuclei from two independent root movies, *n* ≥ 125 nuclei. **d** DII ratiometric signal in the indicated *YUC-TAA1 R2D2* lines in the *pin2* background. Presented are merged confocal images of mDII (red) and DII (yellow). Scale bar = 100 μm. **e** Quantification of relative DII/mDII following 5 μM estradiol and mock treatments of indicated *pin2*-mutant lines. The blue box indicates the region used to quantify the DII/mDII ratio. Shown are means (±SE) of two independent roots, *n* ≥ 300 nuclei. **f** Quantification of root nuclei instantaneous speed following 5 μM estradiol treatment of indicated *pin2*-mutant lines. Data points are concatenated means (±SD) of ~1000 nuclei from two independent root movies. **g-h** Quantification of the root nuclei instantaneous angles in the indicated lines following treatment with 5 μM estradiol or untreated mock control. Data points are concatenated means (±SD) of ~1000 nuclei from two independent root movies. **i-j** Single-cell scatter analysis of velocity at the X and Z-dimensions. Time is indicated by the rainbow scale. $T_0$ is 10 min following mock or estradiol treatment. Ctr indicates controls, heterozygous lines for *promoter-specific:XVE:YUC1:TAA1; R2D2* and the *aux1* (**i**) or the *pin2* (**j**) mutation. *aux1* or *pin2* are also heterozygous for *promoter-specific:XVE:YUC1:TAA1; R2D2* but homozygous for *aux1* or *pin2*. Statistical analysis is presented in Supplementary Fig. 17. For all experiments, the *n* number for each line and time point is indicated at Source data file.

biophysical properties. A recent modeling approach found that carrier-mediated auxin transport alone does not explain the root-tip auxin distribution and showed that plasmodesmata enable fluxes between adjacent tissue layers by allowing auxin to move between the transport streams created by the polar PIN proteins[57]. It would be interesting to test the model with the cell-type-specific auxin synthesis tools generated here to address the importance of local auxin biosynthesis in simplistic auxin redistribution via plasmodesmata. The single-cell tracking approach developed here should allow biologists and modelers from other disciplines to study whole-organism kinetics and morphology of various critical processes.

## Methods

**Plant materials and growth conditions.** All the *Arabidopsis thaliana* plants used in this study were in a Col-0 ecotype background. Seeds were plated on a medium containing 0.5 Murashige-Skoog (MS) medium, 1% sucrose, and 0.8% agar on vertical plates (16 × 16 cm² petri dishes), stratified for 2 days at 4 °C, and then transferred to growth chambers (Percival CU41L5) at 21 °C, 100 mE m² s⁻¹ light intensity under long day light (16 h light/8 h dark). For propagation, plants were grown in growth rooms with a long day light (16 h light/8 h dark) at 21 °C. Seeds used were previously described: *DR5:VENUS*[28], *aux1-7*[58], *pin2* (*eir1* allele)[22], and *35S:H2B-RFP*[36]. *R2D2* lines were generated by transforming Col-0 plants with the *R2D2* vector described previously[51], details are given in the section "Plant genetics".

**Gene cloning.** Cell-type-specific *YUC1* and *TAA1* inducible lines were generated by cloning the *YUC1-2A-TAA1* cassette into XhoI and SpeI sites of the pER8 vector[59]. The full-length cDNA of *YUC1* was cloned into the BamHI site, and the full-length cDNA of *TAA1* was cloned into the BglII site of the pM2A vector containing 2 A peptides[60] to generate the complete *YUC1-2A-TAA1* cassette. For construction of the *YUC1-2A-TAA1-2A-GFP* cassette, *GFP* was cloned into the SmaI site of the pM2A vector containing *YUC1-2A-TAA1*. For the tissue-specific activation of the *YUC1-2A-TAA1* or the *YUC1-2A-TAA1-2A-GFP* cassette, the genomic DNA containing promoters of *SHR*, *SCR*, *APL*, *CLE40*, *WOX5*, or *WER* were used. The primers are listed in Supplementary Table 2. The floral dipping method was used in transformation of *Arabidopsis*. At least ten independent lines were generated, and two of each transformation were obtained and analyzed in T3 homozygous lines.

**Plant genetics.** Inducible cell-type-specific auxin biosynthesis constructs were introduced into *DR5:VENUS* background by transformation. At least ten independent lines were generated, and two of each transformation were obtained and analyzed in T3 homozygous lines. The different inducible cell-type-specific *YUC1-TAA1* inducible and *DR5:VENUS* homozygous lines were crossed with the *35S:H2B-RFP* homozygous line to generate F1 seeds. The F1 seeds were used for data analyses presented in Figs. 2–4.

The R2D2 marker was introduced into the *aux1* (*aux1-7*) and the *pin2* (*eir1* allele) mutant backgrounds. At least ten independent lines were generated, and two of each transformation were obtained and analyzed in T3 homozygous lines. Inducible cell-type-specific *YUC1-TAA1* biosynthesis constructs were introduced into *R2D2 aux1* and *R2D2 pin2* homozygous backgrounds by transformation. At least ten independent lines were generated, and two of each transformation were obtained and analyzed in T3 homozygous lines. Homozygous plants for all three constructs or mutant lines (cell-type-specific *YUC1-TAA1* biosynthesis constructs,

R2D2, and *aux1*) were crossed with *aux1-7* mutant and Col-0 to generate F1 seeds used for data analyses presented in Fig. 6. Similarly, homozygous plants for all three constructs or mutant lines (cell-type-specific *YUC1-TAA1* biosynthesis constructs, R2D2, and *pin2*) were crossed with the *pin2* mutant and Col-0 (as a control) to generate F1 seeds used for data analyses presented in Fig. 6. This allowed us to analyze control plants that have the same transformation event for cell-type-specific *YUC1-TAA1* biosynthesis constructs and the R2D2 marker.

**Root elongation assay.** Seeds were grown on MS plates, stratified at 4 °C for 2 days, grown vertically in long day conditions in a growth chamber for 4 days at 21 °C. Seedlings were then transferred to plates containing 5 μM estradiol and grown vertically for another 7 days before imaging using an HP scanner. Image analyses were performed using Fiji software.

**Chemical application.** Estradiol (20 mM stock solution, dissolved in ethanol), IAA (20 mM stock solution, dissolved in ethanol), NPA (25 mM stock solution, dissolved in DMSO), 1-NOA (20 mM stock solution, dissolved in DMSO), and L-tryptophan (50 mM stock solution, dissolved in doubly distilled H₂0) were applied to the agar medium at the indicated concentrations. Seedlings were placed on agar plates and roots were uniformly supplemented with chemicals. Time points for each experiment are indicated in the figure legends.

**Microscope confocal imaging.** Seedlings were stained in 10 mg L⁻¹ propidium iodide for 2 min and rinsed in water for 30 s. Confocal microscopy was performed using a Zeiss LSM780 inverted confocal microscope equipped with a 20×/0.8 M27 objective lens. GFP and VENUS were excited using an argon-ion laser, whereas tdTomato, RFP, and propidium iodide were excited using a diode laser. Emissions were detected sequentially with ZEN to prevent crosstalk between fluorophores. Excitation and detection of fluorophores were configured in two separate channels. GFP was excited at 488 nm and detected at 498–530 nm. Venus was excited at 514 nm and detected at 493–578 nm for *DR5:VENUS* and at 508–543 nm for *DII-VENUS*. tdTomato was excited at 561 nm and detected at 597–696 nm. Propidium iodide was excited at 561 nm and detected at 578–718 nm. *pWOX5:mCHERRY* was separated from propidium iodide using two tracks to avoid crosstalk with the following filter settings: Track 1: 713 − 748 (PI), Track 2: 578 − 601 (mCHERRY).

**Time-lapse tracking video.** For time-lapse tracking videos, seeds were sown on MS plates, stratified at 4 °C for 2 days, and grown vertically in growth chamber for 4 days at 21 °C. Seedlings were rinsed in water for 30 s and transferred to cell culture dishes with glass bottoms (35 mm, Greiner Bio-one). Seedlings were then covered with 0.7% agar MS with or without 5 μM estradiol. Imaging commenced at 20 min for *DR5:VENUS* and 10 min for R2D2 following initiation of estradiol treatment using a Zeiss LSM 780 inverted microscope (estradiol was present in the media for the whole 6 h movie). To create 4D images of the *DR5:VENUS* marker, 30 slices (Z-stacks: 1.40 μm interval for *DR5:VENUS* movies (Figs. 2–3) and 6.22 μm interval for R2D2 movies (Fig. 6)), 4 tiles were used to image all three developmental zones (20× objective), and images were collected at 40 time points (an image every 8.75 min). For real-time imaging of the R2D2 marker, Z-stacks of 10 slices and 2–3 tiles were used to image root (20× objective), and images were collected at 18 time points (an image every 10 min).

**Quantitative RT-PCR analysis.** Ten-day-old seedlings treated with estradiol (5 μM, 90 min) were collected, and RNA was extracted with the PureLink RNA Mini Kit (Invitrogen). Poly(dT) cDNA was synthesized from 1 μg of total RNA using the High-capacity cDNA Reverse Transcription Kit (ThermoFisher). Expression of selected genes was analyzed using quantitative real-time PCR, ABI Step One Plus System, and ABI software. The qRT-PCR reaction consisted of gene-specific

primers (Supplementary Table 2), Fast SYBR Green Master Mix (Applied Biosystems, Thermo Scientific), and cDNA template. Data were analyzed using the $2^{-\Delta\Delta Ct}$ method[61].

**Statistical analysis and reproducibility**. The two-tailed Student's $t$ test was performed when two groups were compared unless indicated otherwise. Statistical significance was determined at the indicated $p$ values. GraphPad Prism 8.0 software was used for statistical analysis and graphing. All experiments were independently reproduced three times.

**Single-cell tracking**. Single-cell tracking was performed using IMARIS with the Imaris surface mode. The following software parameters were used in all experiments shown in Figs. 2–4: (1) Enable Region Of Interest = false, (2) Enable Region Growing = true, (3) Enable tracking = true, (4) Source Channel Index = 1, (5) Enable Smooth = true, (6) Surface Grain Size = 1.5, (7) Enable Eliminate Background = true, (8) Diameter Of Largest Sphere = 6.23, (9) Enable Automatic Threshold = false, (10) Manual Threshold Value = 20, (11) Active Threshold = true, (12) Enable Automatic Threshold B = true, (13) Manual Threshold Value B = variable, (14) Active Threshold B = false, (15) Region Growing Estimated Diameter = 8.30, (16) Region Growing Background Subtraction = true, (17) "Quality" above 5,000–15,000, (18) "Number of Voxels" above 10,000, (19) Algorithm Name = Autoregressive Motion, (20) Max Distance = 9–15 (automatically set by the software), (21) Max gap Size = 3, (22) "track Duration" above 3,600.000 s.

For experiments using R2D2 marker (Fig. 6), the same software parameters were used with the following modifications: (10) Manual Threshold Value = 26 and (22) "track Duration" above 1,800.000 s.

**Vertical stage microscopy**. Vertical stage microscopy for long-term tracking of root meristems was performed as described[62,63]. Roots were imaged with a vertically positioned LSM700 inverted confocal microscope and Zeiss Zen 2.3 "Black" software with 20× objective. Z-stacks of 30–42 μm were set to ensure that each cell was imaged at least once. For the root-tracking, the TipTracker MATLAB script (Zen Black) was used; interval duration was set between 600 s (10 min) and 720 s (12 min). The resulting images were concatenated and analyzed using ImageJ (NIH; http://rsb.info.nih.gov/ij). For registration, ImageJ macros "correct 3D drift", "StackReg", or "MultiStackReg" were used.

**UV laser ablation**. The UV laser ablation was performed as described[62,64]. In the center of the meristematic zone, two epidermis cells or one stele cell (or one group of stele cells) were ablated and observed at 12 and 24 h after the ablation. Confocal imaging was performed with Zeiss LSM800 inverted microscopes using a 40× objective. Images were analyzed using the ImageJ and Zeiss Zen 2.3 "Black" or "Blue" software. Where necessary, images were processed by adjusting contrast and lightness.

**MATLAB**. An analytical framework designed for automated high-throughput quantification of single-cell migration and morphokinetics was developed. The analysis combined and associated a vast amount of spatiotemporal data across multiple experiments into quantitative measurements. We demonstrated the power of the software by quantifying variations in cell population migration rates while explicitly detecting and quantifying single-cell morphokinetics signatures during collective cell movement. CLSM-acquired 4D (3D and time-lapse images) were subjected to single-cell segmentation and cell tracking using IMARIS and an in-house MATLAB code; 36 morphokinetic features of single-cell and cell population were calculated (Supplementary Table 1).

**Reporting summary**. Further information on research design is available in the Nature Research Reporting Summary linked to this article.

## Data availability
All the data supporting the findings of this study are available within the article, its Supplementary Information files or from the corresponding author upon request. Source data are provided with this paper.

## Code availability
All in house code used in this study us available from the corresponding author upon request

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

## Acknowledgements

This work was supported by grants from the Israel Science Foundation (2378/19 to E.S.), the Joint NSFC-ISF Research Grant (3419/20 to E.S. and Z.D.), the Human Frontier Science Program (HFSP—LIY000540/2020 to E.S.), the European Research Council Starting Grant (757683- RobustHormoneTrans to E.S.), PBC postdoctoral fellowships (to Y.H. and M.O.), NIH (GM114660 to Y.Z.), Breast Cancer Research Foundation (BCRF to I.T.).

## Author contributions

This scientific study was conceived and planned by Y.H., M.O., I.T., and E.S. Experiments were performed by Y.H. Cell-type-specific auxin inducible *Arabidopsis* lines were generated by M.O. The laser ablation experiments and long-term vertical imaging microscopy were performed by L.H. MATLAB single-cell image analysis tools were developed by O.D., O.M., O.C., and I.T. Cell-type-specific auxin constructs were cloned by Y.H. and Q.C. The work was designed and supervised by Z.D., J.F., Y.Z., I.T., and E.S. The paper was written by Y.H. and E.S.

## Competing interests

The authors declare no competing interests.
