## [Peer Review File · Nature Communications]

REVIEWER COMMENTS

Reviewer #1 (Remarks to the Author):

This manuscript describes nicely the use of a microscopy based single cell trajectory tracing to better characterize root cell behavior during growth and in response to auxin.

This approach consists on time laps tracking of the cell nuclei and a MATLAB based algorithm to record multiple nuclei behavior.

While this approach looks interesting and very promising for so many applications, I am however a bit puzzled by the results and their interpretation.

The first part of the manuscript describes the use of an inducible system to promote cell type specific auxin biosynthesis, a very interesting tool that unfortunately is not giving that exciting results to my point of view.

To fully describe the specificity of the different inducible lines generated, the same analysis of nuclei behavior should have been conducted also on root treated with different concentration of auxin, but also with NPA as well as with 1-NOA. Then compare these profiles to the line phenotypes to better understand their difference. To me the amount of auxin produced by each construct have more impact on root growth than the auxin biosynthesis localization itself.

Secondary, what you define as skewing in your analysis is not convincing enough. On Fig2, your X and Y velocity may be only due to asynchronous cell divisions in the meristematic zone (Fig2), rather than real skewing. In the past, skewing was defined when the general direction of the root growth deviates from a gravity vector throughout its development (even though skewing happens with no gravity). This is usually recorded after multiple hours or few days of growth.

The problem is that the videos are recorded on horizontal microscope (if I am not mistaken and as it is not fully described in the M&M), then the gravity vector is perpendicular of the growth and roots don't behave as "normal" therefore the skewing hypothesis for me cannot be fully supported. If you want to fully study this phenomena, some skewing related mutants such as spr1 or sku5 should be analyzed as well so show how skewing can be displayed using your method.

In general, this manuscript despite using very promising approaches, do not lead to a major discovery on the role of auxin or on meristematic cells behavior, to be published in Nature Communication.

More specific comment on the manuscript:

-Auxin production in certain cell types significantly affects root growth.

Great

-Single-cell nucleus tracking approach determined the morphokinetics of Arabidopsis root growth and tip skewing.

In general, this figure 2 does not give enough info on which root parts are shown on the graph. Where is the QC, Meristematic Zone(MZ), Elongation Zone(EZ), etc... from your microscopic pictures on the graphs. Does it correlate with the velocities?

Fig2 D and suppl Fig7&11, why so many nuclei trajectories are as hectic and seems to cross from one side of the root toward the other side?

Especially in the elongation zone some cells show horizontal trajectories. Is it a program glitch or is it true? Please if true support with a close-up video.

When looking at your videos (very nice btw) the cells don't seem to behave like that.

If the program actually miss-track certain nuclei, then you need to precise the % of wrong trajectories.

L103: is it not the area around the QC that shows the more X and Z velocities? Then these velocities would be due to division rather than skewing. Please precise.
If it is skewing, I would expect the velocities of cells to be directed toward the same direction on the X or on the Z axis. This information is missing. If not it is just random movement due to division.

L109: please don't use "root tip" be more precise (MZ,EL,MaturationZ)

L114: "Surprisingly" not maybe the right term, in the meristem cell movement is mostly due to division and not elongation. These divisions have been shown by multiple ways are not synchronous, therefore the lack of coordination.

L116: Supp Fig5 not the right one to support the skewing I guess. Video 4 ok-ish but difficult to see. You cannot say on the video if it is the meristem zone or the elongation zone responsible for the skewing.

-Cell-type specific auxin production differentially affects root kinetics and skewing as shown by single-cell image analysis.

This part shows (too) many parameters analyzed by your new method but it greatly lacks the biological explanations (ie: is it interesting/useful to put the velocity and the instantaneous speed graphs? They look the same and in term of biology I don't see the relevance of having both, please prove me wrong if I haven't understood it).

Many of the parameters are not explained, like I cannot understand the interest to look at MSD, even by looking at definition in the supplementary what does it means in term of biology.

What does "area" means, even with the very sparse definition a reader have no clue to understand what it means, where do you measure the parameters A.B.C and R ?

What is the Eccentricity ?! The definition is deeply obscure, what does it means in biology please?

Please display the graphs as they are cited in the text too to facilitate the reader.

L135 "directional change" graph is not present in Fig.3

Finally, for which part of the root is the measures taken? The control shows almost no coordinated motility, while on Fig2 the entire elongation zone displays a great coordination.

I have the feeling you would have get the same results if you had placed the root in different concentration of auxin. I bet the APL line looks like ctr because the level of extra produced auxin is very low.

That would be a control to do and it would be cool to show that auxin treatment may differs from your different lines (PNA and NOA-1 too).

L154 "stele" not endodermis

L159 induction in "Stele" not endodermis

L163 & Fig4.

Controls, auxin, NPA, 1-NOA would be very informative to better understand what the different lines are doing.

The thing is that all the induced lines shows an inhibition of root growth just like any exogenous auxin treatment. Is the increase of coordinated motility not just a consequence of growth inhibition? For me very little to do with skewing.

L167 I don't see any skewing on videos

Fig4: PCLE40 glitch on the top dots at the beginning of the time

-Mapping IAA movement in the root

Super cool approach but it is only confirming (with maybe a bit more precisions) what was already published.

L191 is auxin moving and accumulating or just locally synthesized? (the stele ablation answers but only partially to this question)

Please add experiments with NPA and 1-NOA

-PIN2 and AUX1 are required for directional auxin movement and root growth but not root skewing. Confirming previous experiment of the part above and published data.

Fig6 how are the sun-plots made is very unclear, the axis represents the velocities, how can a velocity be negative? These graphs are very difficult to understand and need to be clarified before reader can draw its own conclusions.

SupplFig11 mock con and aux1, almost none of the cells move vertically, they move more horizontally, please add a video to prove that as it seems a peculiar behavior.

Minor points:

Why is the control called "con", universal abbreviations are "ctr" or "ctrl", it just makes a bit difficult to understand the figures at first sight.

Primers missing TAA1

Reviewer #2 (Remarks to the Author):

This manuscript by Hu et al. describes a novel approach to deduce the direction and extent of auxin transport in the Arabidopsis root meristem. The approach is original and "high-tech", and in my opinion the main value of the paper is the sound demonstration that root development can be observed in such high 4D resolution as described, and that the observations confirm a number of previous observations or notions, notably the "reverse fountain" model. Beyond, the paper also reports root skewing as a potential, overlooked auxin response. There are few comments or suggestions for improvement from my side, but I believe the following additions/corrections should be taken into account in a revised version and would make a stronger paper:

- An interesting observation is the uncoupling of root skewing from other, clearly auxin-dependent tropisms. Root skewing can be interpreted as a sort of "search" mechanism, and in the context of tissue culture, it has been interpreted as a growth response that results from the "conflict" between substrate penetration and gravity. This made me wonder whether skewing is at all observed when roots are growing straight into medium? For example, if roots are surrounded by liquid medium, such as in the tracking set up published by the Friml lab, does skewing still occur?

- Among the transgenic auxin induction lines, the pAPL lines behave differently, possibly because they respond the least to the estradiol induction. The authors state repeatedly (lines, 61, 237) that pAPL confers expression in companion cells, but I wonder whether this is indeed correct? For sure, pAPL expresses during the final stages of sieve element differentiation. And possibly, thereafter, in phloem-associated pericycle cells. The authors should carefully check the literature to verify this.

- In this context, it is perhaps noteworthy that a very recent publication (Moret et al. Nature Plants) has shown that differentiating root tip phloem sieve elements already seem to produce an

auxin boost through YUCCA expression, in the same cells that express APL. This paper also suggests that auxin transport through the phloem is well buffered, which may explain the absence/weakness of effects in the present manuscript.

- Irrespectively, this raises a more general concern: that is, is auxin production limited in any of the scenarios? One assumption behind the approach is that the primary substrate, tryptophan, is available at non-limiting levels, which might not be the case. One simple control experiment would be to treat the lines with estradiol in the presence or absence of simultaneously added tryptophan and/or indole-3-pyruvic acid and check whether there are any differences?

- Coming back to the skewing, it is remarkable that the pAPL line is the only one that does not respond. It makes me wonder whether this is due to the fact that it is the only line in which auxin production would be induced in an asymmetric manner? The other lines confer more or less concentric induction of auxin production around the respective root circumference, and their comparatively high induction might just swamp the root with auxin, unlike in the pAPL line. Thus, is the skewing perhaps related to the radial symmetry breaking in the vasculature? Does pAPL not abolish skewing because asymmetric auxin induction in the phloem does not break auxin production asymmetry? Maybe this can be answered from the data. That is, is auxin response becoming more or less concentric as indicated by the DR5/R2D2 markers upon induction in the different lines? Or, can skewing be abolished by auxin "flooding" in a liquid set up (if the roots still skew there, see above), but not by unilateral application?

- There are a number of typos and grammatical oddities throughout the manuscript, for example the very first sentence of the introduction is incomprehensible.

In the enclosed manuscript, we performed new experiments that addressed all concerns raised by the reviewers, that significantly improved the manuscript. Below is a detailed response to all comments (in blue font).

REVIEWER COMMENTS

Reviewer #1 (Remarks to the Author):

This manuscript describes nicely the use of a microscopy based single cell trajectory tracing to better characterize root cell behavior during growth and in response to auxin. This approach consists on time laps tracking of the cell nuclei and a MATLAB based algorithm to record multiple nuclei behavior.

While this approach looks interesting and very promising for so many applications, I am however a bit puzzled by the results and their interpretation.

The first part of the manuscript describes the use of an inducible system to promote cell type specific auxin biosynthesis, a very interesting tool that unfortunately is not giving that exciting results to my point of view.

To fully describe the specificity of the different inducible lines generated, the same analysis of nuclei behavior should have been conducted also on root treated with different concentration of auxin, but also with NPA as well as with 1-NOA. Then compare these profiles to the line phenotypes to better understand their difference. To me the amount of auxin produced by each construct have more impact on root growth than the auxin biosynthesis localization itself.

We have carried the suggested experiments and analyzed the root's response to increasing IAA concentrations (new Sup. Fig. 10). The results showed the expected dose (concentration) dependent responses. Interestingly, root tip skewing was rapidly inhibited already in response to 20 nM IAA treatment. We added these new results and discussion in the text and compared them to the cell-type auxin synthesis responses. We agree with the reviewer that the amount of auxin produced by the cell-type-specific promoters profoundly affects the response. We modified the text to emphasize this point. In addition, the results suggest that the responses, in addition to auxin levels, are determined by auxin movement. The auxin reporters indicate that its rapid movement buffers the spatial synthesis of auxin. *pWOX5* is likely the best example as auxin is rapidly moving shootwards to mediate much of the growth inhibition. The rapid but transient inhibition in cell velocity in response to 20 nM IAA treatment points to our monitoring system's high spatiotemporal sensitivity and suggest that *pAPL* non-responsiveness is not likely a result of insufficient activation of IAA but rather reflects the phloem buffering in IAA response, transport or metabolize.

As suggested, we generated root single-cell image analysis treated with NPA or 1-NOA, with and without *pWOX5* dependent auxin induction (new Sup. Fig. 13). The data further demonstrates how 1-NOA to a large extent, and NPA to a lower but significant extent, delays root growth inhibition generated by QC-specific auxin induction. Interestingly, 1-NOA can partially repress the *pWOX5* dependent auxin induction root skewing inhibition. We added these results (new Sup Fig. 13) and discussed them in the text.

Secondary, what you define as skewing in your analysis is not convincing enough. On Fig2, your X and Y velocity may be only due to asynchronous cell divisions in the meristematic zone (Fig2), rather than real skewing. In the past, skewing was defined when the general direction of the root growth deviates from a gravity vector throughout its development (even though skewing happens with no gravity). This is usually recorded after multiple hours or few days of growth.

The problem is that the videos are recorded on horizontal microscope (if I am not mistaken and as it is not fully described in the M&M), then the gravity vector is perpendicular of the growth and roots don't behave as "normal" therefore the skewing hypothesis for me cannot be fully supported.

We monitored root skewing again, with and without NAA treatment, using a long-term vertical stage microscope in a 25 hours' time course, coupled with laser ablation to mark the cells (new Movie S5). The roots show evident skewing in the meristem zone, which is inhibited following NAA application.

In addition, root skewing of *pWOX5:YUC1-TAA1; DR5:VENUS* and *pSCR:XVE:YUC1-TAA1; DR5:VENUS*, with and without estradiol induction, was recorded on a long-term vertical stage microscope (new Movie S7). The new results clearly show root skewing that takes place only in non-treated estradiol seedlings. Therefore, root skewing is conserved and reproducible in horizontal and vertical stage microscopes using long-term vertical stage microscope or single-cell tracking.

If you want to fully study this phenomena, some skewing related mutants such as *spr1* or *sku5* should be analyzed as well so show how skewing can be displayed using your method.

To monitor skewing using the single-cell system we developed, one must introduce *35S:H2B-RFP* into the *spr1* and *sku5* backgrounds and select homozygous lines. We have obtained the *spr1* and *sku5* mutants and crossed them to *35S:H2B-RFP*. However, we need ~7 more months to generate the homozygous lines and complete this experiment.

We believe that the new videos and images we generated, using independent approaches, showing evident root skewing using the long-term vertical-stage microscope, with and without cell-type-specific auxin induction, is sufficient to address these concern.

In general, this manuscript despite using very promising approaches, do not lead to a major discovery on the role of auxin or on meristematic cells behavior, to be published in Nature Communication.

More specific comment on the manuscript:

-Auxin production in certain cell types significantly affects root growth. Great-Single-cell nucleus tracking approach determined the morphokinetics of Arabidopsis root growth and tip skewing.

In general, this figure 2 does not give enough info on which root parts are shown on the graph.

Where is the QC, Meristematic Zone(MZ), Elongation Zone(EZ), etc... from your microscopic pictures on the graphs. Does it correlate with the velocities?

We have now marked the different regions on the graph (Fig. 2) to make this clearer.

Does it correlate with the velocities?

Yes, the different zones perfectly correlate with velocities. We revised the graph and legends to make this clearer (Fig. 2).

Fig2 D and suppl Fig7&11, why so many nuclei trajectories are as hectic and seems to cross from one side of the root toward the other side?

Especially in the elongation zone some cells show horizontal trajectories. Is it a program glitch or is it true? Please if true support with a close-up video. When looking at your videos (very nice btw) the cells don't seem to behave like that. If the program actually miss-track certain nuclei, then you need to precise the % of wrong trajectories.

We added a trajectory movie of single-cell tracking (new Movie 6). The meristem zone trajectories seem hectic with very poor coordination motility because opposing cells skew to opposing directions as the root grows. In the elongation zone, the cells are traced primarily in the vertical axis (Y-axis dimension). There are indeed a few cells that show horizontal tracing (4.5% of the cells). These are primarily a result of captured meristem-zone cells (it is challenging to separate between the two zones in live videos).

L103: is it not the area around the QC that shows the more X and Z velocities? Then these velocities would be due to division rather than skewing. Please precise. If it is skewing, I would expect the velocities of cells to be directed toward the same direction on the X or on the Z axis. This information is missing. If not it is just random movement due to division.

Cells around the QC show relatively high X and Z-axis velocities. These results are in-line with root skewing. The velocities are not random but rather highly coordinated as cells on opposite sides of the root rotate in opposite directions, reflected by an increase

in X and Z axis velocities, reduction in Y-axis, and reduction in coordinated motility. Our measurements show that in a 5.5-hour movie, only 25% of the cell divide. However, in the meristem, we detect cells' skewing motion in opposing direction, correlating with their position. In addition, the tracking shows that movement is not a single layer shift, as expected from cell-division, but rather a curve motion over the entire root (new Movie S6). We modified the text to make the point clearer. Finally, we show that roots treated with auxin, although slightly reduced, keep dividing in the meristem zone (new Sup. Fig. 8). However, the region's cells' coordinated motility is highly associated, suggesting against cell-division as the main factor explaining low coordinated motility in the meristem. Altogether, we show by live imaging root meristem skewing and provide further evidence that the meristem's low coordinated motility is likely reflecting root skewing.

L109: please don't use "root tip" be more precise (MZ,EL,MaturationZ)

We modified the text accordingly.

L114: "Surprisingly" not maybe the right term, in the meristem cell movement is mostly due to division and not elongation. These divisions have been shown by multiple ways are not synchronous, therefore the lack of coordination.

We generated new movies using single-cell tracking and long-term vertical-stage microscopy. Both approaches present evident root skewing.

We tested the frequency of cell division in the meristematic zone in a 20-hour vertical-stage microscope experiment comparing mock and NAA treatment (new Sup. Fig. 8). NAA treatment showed only a mild inhibition in cell-division under these settings. However, root coordinated motility in these conditions is very high, thus likely reflecting root skewing and not cell-division.

The reduction in the Y-axis cell-velocity and the increase in the X and Z-axis are explained by skewing vectorial motion and not cell division.

The word surprisingly was removed and the possible involvement of cell-division is discussed.

L116: Supp Fig5 not the right one to support the skewing I guess. Video 4 ok-ish but difficult to see. You cannot say on the video if it is the meristem zone or the elongation zone responsible for the skewing.

We generated a new video recording root skewing (with or without NAA) on a vertical microscope stage in a 25 hours' time course (see Movie S5). The video clearly shows root skewing in the meristematic zone. In addition, we now show root skewing videos of *pWOX5:XVE:YUC1-TAA1*; *DR5: VENUS* and *pSCR:XVE:YUC1-TAA1*; *DR5: VENUS* on a vertical microscope (see Movie S7). The results from these new movies and the single-cell tracking approach strongly support the meristematic zone skewing.

-Cell-type specific auxin production differentially affects root kinetics and skewing as shown by single-cell image analysis.

This part shows (too) many parameters analyzed by your new method but it greatly lacks the biological explanations (ie: is it interesting/useful to put the velocity and the instantaneous speed graphs? They look the same and in term of biology I don't see the relevance of having both, please prove me wrong If I haven't understood it).

We removed the instantaneous speed and MSD parameters out of Fig. 3.

Many of the parameters are not explained, like I cannot understand the interest to look at MSD, even by looking at definition in the supplementary what does it means in term of biology.

What does "area" means, even with the very sparse definition a reader have no clue to understand what it means, where do you measure the parameters A.B.C and R ? What is the Eccentricity ?! The definition is deeply obscure, what does it means in biology please?

We revised the text to make this clearer. This appears both in the text and Fig. 3 legends. A major aim of this manuscript is to present the comprehensive capacities and strength of the approach we developed. We think it's important to include parameters that did not present significant changes following cell-type-specific auxin induction, such as area (nuclei area measured for single cells) and eccentricity (ellipsoid shape deviating from circular). We believe that the single-cell kinetics approach we present here will allow the whole community to advance the morphokinetics spatiotemporal characteristics to reveal broad and dynamic developmental processes.

Please display the graphs as they are cited in the text too to facilitate the reader.

We revised the text and graphs accordingly.

L135 "directional change" graph is not present in Fig.3

We corrected the text.

Finally, for which part of the root is the measures taken? The control shows almost no coordinated motility, while on Fig2 the entire elongation zone displays a great coordination.

The elongation zone in Figure 2 nicely shows high coordinated motility as cells are elongating primarily in the Y axis while the meristem zone is skewing, therefor opposing cells are moving in opposing direction, creating low coordination motility. The data in Figure 3 presents the average parameters value of all nuclei over time, collected for the entire root that we imaged (around 1,000 nuclei from the meristem, elongation, and

maturation zones). Thus, cells of MZ and DZ (more than 67% of cells) in the control and *pAPL* show very low coordinated motility compared to other inducible lines (Fig. 4). We modified the figure legends to make the point more straightforward.

I have the feeling you would have get the same results if you had placed the root in different concentration of auxin. I bet the APL line looks like ctr because the level of extra produced auxin is very low.

That would be a control to do and it would be cool to show that auxin treatment may differs from your different lines (PNA and NOA-1 too).

Indeed, producing auxin within the cells and providing exogenous auxin treatments result in similar patterns. However, the inducible cell-type IAA synthesis allows to investigate the response and movement in a spatial resolution. Roots treated with very low IAA concentrations (20 nM of IAA) showed a significant but transient growth inhibition that was not detected for *pAPL:YUC1-TAA1* (new Sup. Fig. 10). Since *pAPL:YUC1-TAA1* show 4.5-fold activation in *YUC* expression, this would suggest that, on top of the low levels of auxin produced by the *APL* promoter, the phloem miss-responsiveness is also reflecting a buffered response. The importance of auxin level is now better emphasized in the text.

L154 “stele” not endodermis

We have corrected this error

L159 induction in “Stele” not endodermis

We have corrected this error

L163 & Fig4.

Controls, auxin, NPA, 1-NOA would be very informative to better understand what the different lines are doing.

The effect of increasing concentrations of IAA or NPA and 1-NOA treatments has been investigated (new Sup. Fig 10 and 13) and discussed in the text. See comments above.

The thing is that all the induced lines shows an inhibition of root growth just like any exogenous auxin treatment. Is the increase of coordinated motility not just a consequence of growth inhibition? For me very little to do with skewing.

Coordinated motility is not affected by the velocity but rather vectorial motion. Therefore, the increase in coordination motility of the meristem zone cells following auxin induction is not a result of simply growing slower but rather a change in directional motion compared to the neighboring cells. The new data added here of roots videos treated with 20 nM IAA suggest that the processes of root skewing is not fully linked to root growth, as three hours following 20 nM IAA treatment, the roots accelerate, reflected by

an increase in the Y-axis velocity, however the skewing, reflected by coordinated motility remain repressed.

L167 I don't see any skewing on videos

It is impossible to detect root skewing looking at "regular" root growth movies. We generated new movies using the long-term vertical stage microscope, combined with laser ablation, that clearly show root skewing (Movies S4 and S5). We also carried long-term vertical stage microscopy movies of cell-type-specific auxin synthesis roots treated with estradiol. The data show evident skewing in mock-treated roots and inhibition in skewing in estradiol-induced lines (new Movie S7). In addition, we generated new movies of single-cell tracking that clearly show root skewing in the meristem zone (new Movie S6).

Fig4: PCLE40 glitch on the top dots at the beginning of the time

We removed the noise caused by nuclei tracking outside of the root in the CLE40 movie and generated new a Matlab figure (Fig. 4).

-Mapping IAA movement in the root

Super cool approach but it is only confirming (with maybe a bit more precisions) what was already published.

One of the aims of this study was to challenge the current model of reverse fountain IAA transport in the root. The model mainly relied on the transporters subcellular localization and there is limited work supporting the model from the molecule (IAA) point of view. The work presented here, using novel genetic and image analysis tools, indeed supports much of the suggested model. We believe that challenging the model and supporting it, is highly important. In addition, this work present multiple novel evidences. For example, the auxin reflux into the vasculature in an NPA dependent manner which was not established previously (Fig. 5).

L191 is auxin moving and accumulating or just locally synthesized? (the stele ablation answers but only partially to this question)

Please add experiments with NPA and 1-NOA

We generated new movies testing single-cell root response following *pWOX5:XVE:YUC1-TAA1* estradiol induction, with and without NPA or 1-NOA treatment. The results showed that 1-NOA to a large extent, and NPA to a lower but significant extent, have delayed root growth inhibition generated by QC-specific auxin induction. We added these results (new Sup Fig. 13) and discussed them in the text.

-PIN2 and AUX1 are required for directional auxin movement and root growth but not root skewing.

Confirming previous experiment of the part above and published data.

PIN2 and AUX1 were initially identified as mutants affecting root tip rotation (Okada et al 1990, Science). Therefore, the single-cell image analysis results presented here, shed new light and are highly important for the field. As far as we are aware of, prior to this study, root skewing was not observed in live imaging, and single-cell kinetics was not recorded over time. Future research is required to understand the molecular mechanism behind root skewing. The characterization of PIN2 and AUX1 with respect to this phenomenon is an important start.

Fig6 how are the sun-plots made is very unclear, the axis represents the velocities, how can a velocity be negative? These graphs are very difficult to understand and need to be clarified before reader can draw its own conclusions.

In this sun-plot presentation, two cells moving toward opposing directions are reflecting positive/negative velocities. We have clarified the text to make this clearer.

SupplFig11 mock con and aux1, almost none of the cells move vertically, they move more horizontally, please add a video to prove that as it seems a peculiar behavior.

One of the main finding in the manuscript is that cells in the meristem zone largely move in a dynamic motion as a result root skewing. We demonstrate this using single-cell tracking horizontal microscope and using laser-ablation coupled with bright-field vertical confocal microscopy. We added new movies tracking the root tip using both approaches and the results are highly consistent (new Movies S4-7).

Minor points:

Why is the control called "con", universal abbreviations are "ctr" or "ctrl", it just makes a bit difficult to understand the figures at first sight.

"Con" has been changed to "Ctr" throughout the manuscript.

Primers missing TAA1

Primers describing *TAA1* cloning (*YUC1-2A-TAA1* cassette) are described in the methods.

Reviewer #2 (Remarks to the Author):

This manuscript by Hu et al. describes a novel approach to deduce the direction and extent of auxin transport in the *Arabidopsis* root meristem. The approach is original and "high-tech", and in my opinion the main value of the paper is the sound demonstration that root development can be observed in such high 4D resolution as described, and that the observations confirm a number of previous observations or notions, notably the "reverse fountain" model. Beyond, the paper also reports root skewing as a potential, overlooked auxin response. There are few comments or suggestions for improvement from my side, but I believe the following additions/corrections should be taken into account in a revised version and would make a stronger paper:

- An interesting observation is the uncoupling of root skewing from other, clearly auxin-dependent tropisms. Root skewing can be interpreted as a sort of "search" mechanism, and in the context of tissue culture, it has been interpreted as a growth response that results from the "conflict" between substrate penetration and gravity. This made me wonder whether skewing is at all observed when roots are growing straight into medium? For example, if roots are surrounded by liquid medium, such as in the tracking set up published by the Friml lab, does skewing still occur?

We generated new movies to address this issue further. The movies recorded root skewing (with or without NAA) using a vertical microscope in a 25 'hours' time course (new Movie S5). Using laser ablation to mark a specific cell, the video shows evident root meristem skewing. The root tip skewing is NAA dependent.

In addition, we monitored root skewing using the vertical microscope of *pWOX5:XVE:YUC1-TAA1; DR5:VENUS* and *pSCR:XVE:YUC1-TAA1; DR5:VENUS* following estradiol induction (see Movie S7). The results show apparent root skewing that was inhibited by auxin cell-type production.

To further address the reviewers concern, we tested root skewing growing in different agar media. Roots mounted in 0.5% (half-liquid) and 0.7% MS-agar showed very similar skewing pattern over time with no clear effect of the medium (new Sup. Fig. 7).

- Among the transgenic auxin induction lines, the pAPL lines behave differently, possibly because they respond the least to the estradiol induction. The authors state repeatedly (lines, 61, 237) that pAPL confers expression in companion cells, but I wonder whether this is indeed correct? For sure, pAPL expresses during the final stages of sieve element differentiation. And possibly, thereafter, in phloem-associated pericycle cells. The authors should carefully check the literature to verify this.

We thank the reviewer for this important correction. *APL* is expressed throughout the vascular strands in *Arabidopsis* seedlings in a phloem-specific manner. In the root, *APL* is first expressed explicitly in the developing protophloem sieve elements and then expressed slightly higher up in the companion cells and metaphloem sieve elements (*Bonke et al., Naure 2003*). We have corrected the text accordingly.

- In this context, it is perhaps noteworthy that a very recent publication (Moret et al. Nature Plants) has shown that differentiating root tip phloem sieve elements already seem to produce an auxin boost through YUCCA expression, in the same cells that express APL. This paper also suggests that auxin transport through the phloem is well buffered, which may explain the absence/weakness of effects in the present manuscript.

We have addressed this new publication and cited the paper (Moret et al., Nature Comm, 2020).

- Irrespectively, this raises a more general concern: that is, is auxin production limited in any of the scenarios? One assumption behind the approach is that the primary substrate, tryptophan, is available at non-limiting levels, which might not be the case. One simple control experiment would be to treat the lines with estradiol in the presence or absence of simultaneously added tryptophan and/or indole-3-pyruvic acid and check whether there are any differences?

We added new results testing the effect of L-Tryptophan on cell-type-specific IAA mediated root growth. In general, applying different concentration of tryptophan (0, 10, 50 and 100 μ M) did not affect the auxin-mediated root growth inhibition. Only a high concentration of tryptophan (100 μ M) showed a slight inhibition in root growth of *pAPL* and *pSCR* lines (new Sup. Fig. 3). It is therefore possible that *pAPL* weak phenotype is at least in part, driven by the lack of tryptophan substrate, but since the response was relatively weak compared to the other promoters, it is likely that other factors such as transport, metabolism and perception are buffering the response.

- Coming back to the skewing, it is remarkable that the *pAPL* line is the only one that does not respond. It makes me wonder whether this is due to the fact that it is the only line in which auxin production would be induced in an asymmetric manner? The other lines confer more or less concentric induction of auxin production around the respective root circumference, and their comparatively high induction might just swamp the root with auxin, unlike in the *pAPL* line. Thus, is the skewing perhaps related to the radial symmetry breaking in the vasculature? Does *pAPL* not abolish skewing because asymmetric auxin induction in the phloem does not break auxin production asymmetry? Maybe this can be answered from the data. That is, is auxin response becoming more or less concentric as indicated by the DR5/R2D2 markers upon induction in the different lines? Or, can skewing be abolished by auxin "flooding" in a liquid set up (if the roots still skew there, see above), but not by unilateral application?

At this stage, it is not entirely clear if un-concentric auxin induction may affect skewing inhibition. Our new results show that both root skewing and growth are rapidly inhibited in response to low IAA treatment (20 nM) (new Sup Fig. 10). The fact that *pAPL:XVE:YUC1-TAA1* roots show almost undetectable responses to estradiol in the first 5h, although 4-fold change induction in *YUC1* expression, may suggest that the response, both root growth inhibition and skewing, is likely buffered.

- There are a number of typos and grammatical oddities throughout the manuscript, for example the very first sentence of the introduction is incomprehensible.

We have revised the text to correct errors and grammar oddities (these changes are not marked in gray).

REVIEWER COMMENTS

Reviewer #1 (Remarks to the Author):

The authors have answered many of the previous technical concerns, explained intelligently their point of view, have performed many additional experiments and control conditions.

Important point: The title single cell is really misleading as "single cell" usually refers to single cell RNA seq.

First it is more single nuclei tracking and not single cell, hence it would be good to replace this term by something else (ie: unique cell, cell by cell etc...)

I still have a very hard time understanding conceptually some of the results:

I trust the results you provide but I think the interpretation as skewing is not right.

During root skewing we imagine very well the root moving randomly in X and Z in your conditions (Y is just the growth right). But how is it possible that a unique cell goes in a different direction than the others (as you say no coordination)? All cells are in their cell walls and cannot move around independently. I probably have not fully understood the way you measure cell coordination but conceptually how can cells in a cell wall be moving independently in different direction? As you showed nicely that cell division has very little to do with the meristem lack of coordination, how the cells moves? I can imagine that local cell wall elongation could be part of the answer but if it is not a coordinated movement toward one direction is that really skewing or is that just random growth?

It is not root skewing if only a few cells move in another direction than the gravity axis.

Fig4: how can you have an increase of coordinated mobility if you have root growth arrest? I just don't understand conceptually.

Fig 6i,j:

The sunplots of pWER in fig 6j are exactly the same. Is there a mistake?

For both figs (I,j) I still don't understand how these plots are made. How 2 cells can move in opposite directions in X and or Z? I thought the velocity was calculated according to an axis. Here you say the velocities are calculated relatively to different cells?

Personally, this representation is really difficult to read and to interpret by eye. You claim that ctr and aux1 behave similarly, but in fig 6i, aux1 dots are more spread than ctr. How do you calculate that 2 sunplots profiles are significantly different? This look very fancy, but I don't see how it helps the story. It makes it even more difficult to interpret

In Suppl 16 pSHR ctr and aux1 look really similar (using Y and X velocity), are they significantly different like the one on fig 6i pSHR?

I don't think these results prove anything, but that auxin induction reduces root growth and that is not aux1 or pin2 dependent (already published). The skewing arrest is just a consequence of root growth arrest.

I however agree with you that what you call skewing is not aux1 or pin2 dependent.

Maybe I just haven't understood anything, so I let the editor read my points and decide if my concerns are legitimate or just stupid.

Minor points:

"This is supported by the rapid suppression in meristem zone of coordinated motility in response to

low- and high-concentration IAA treatments (Sup. Fig. 10a).”
Here I think you wanted to say the rapid augmentation/increase

“Primers missing TAA1
Primers describing TAA1 cloning (YUC1-2A-TAA1 cassette) are described in the methods.”

No. in the M&M, you cite ref 59 and 60 but these 2 refs don't use YUC1 and TAA1. They don't provide the primers you used. Plus, even if they did, it doesn't cost you anything to put them in the suppl table 2 as the other primers. It helps the reader that want to use your constructs one day. In many animal journals you would have to even add the full sequences as well as the full annotated plasmids.
So please add these primers.

Reviewer #2 (Remarks to the Author):

The authors have comprehensively and adequately responded to my comments, from my side the manuscript is ready to be published.

We are grateful to the reviewers for their comments. In the enclosed manuscript, we addressed the concerns raised by reviewer 1. Below is a detailed response to his comments (in blue font). Changes in the text are marked in gray.

REVIEWER COMMENTS

Reviewer #1 (Remarks to the Author):

The authors have answered many of the previous technical concerns, explained intelligently their point of view, have performed many additional experiments and control conditions.

Important point: The title single cell is really misleading as "single cell" usually refers to single cell RNA seq.

First it is more single nuclei tracking and not single cell, hence it would be good to replace this term by something else (ie: unique cell, cell by cell etc...)

We modified the title: "Cell kinetics of auxin transport and activity in *Arabidopsis* root growth and skewing".

I still have a very hard time understanding conceptually some of the results: I trust the results you provide but I think the interpretation as skewing is not right. During root skewing we imagine very well the root moving randomly in X and Z in your conditions (Y is just the growth right). But how is it possible that a unique cell goes in a different direction than the others (as you say no coordination)? All cells are in their cell walls and cannot move around independently. I probably have not fully understood the way you measure cell coordination but conceptually how can cells in a cell wall be moving independently in different direction? As you showed nicely that cell division has very little to do with the meristem lack of coordination, how the cells moves? I can imagine that local cell wall elongation could be part of the answer but if it is not a coordinated movement toward one direction is that really skewing or is that just random growth?

It is not root skewing if only a few cells move in another direction than the gravity axis.

Direct neighboring cells do not move in different directions. However, in the meristem, cells move in opposite directions when localized at opposing sides of the root. We marked

all the cells within a 200- μ m radius from a chosen center cell in the coordinated motility calculation. For every cell in the perimeter, we calculated the cosines of the velocity angles in relationship to the selected center cell. That means that the comparison is not testing coordination between direct neighboring cells but rather the whole meristem region. To clarify this point, we introduced a new 3D illustration (Figure 2e).

Fig4: how can you have an increase of coordinated mobility if you have root growth arrest? I just don't understand conceptually.

Upon IAA syntheses, the meristem stop skewing, reflected by low X and Z-dimension movement. The primary motion comes from the Y-dimension, therefore showing high coordinated motility. For example, when two cells are positioned on opposite sides of the meristem epidermis layer, while skewing, they move at opposite X and Z-dimension. But when skewing is inhibited, these two cells placed on opposite sides of the meristem move in one dimension only (see new illustration in Figure 2e).

Fig 6i,j: The sunplots of pWER in fig 6j are exactly the same. Is there a mistake? For both figs (i,j) I still don't understand how these plots are made. How 2 cells can move in opposite directions in X and or Z? I thought the velocity was calculated according to an axis. Here you say the velocities are calculated relatively to different cells?

Personally, this representation is really difficult to read and to interpret by eye. You claim that ctr and aux1 behave similarly, but in fig 6i, aux1 dots are more spread than ctr. How do you calculate that 2 sunplots profiles are significantly different? This look very fancy, but I don't see how it helps the story. It makes it even more difficult to interpret In Suppl 16 pSHR ctr and aux1 look really similar (using Y and X velocity), are they significantly different like the one on fig 6i pSHR?

I don't think these results prove anything, but that auxin induction reduces root growth and that is not aux1 or pin2 dependent (already published). The skewing arrest is just a consequence of root growth arrest.

I however agree with you that what you call skewing is not aux1 or pin2 dependent. Maybe I just haven't understood anything, so I let the editor read my points and decide if my concerns are legitimate or just stupid.

We replaced the sun-plot graphs in Figure 6i with an absolute scatter plot (Figure 6i). We also replaced the sun plot in Sup 16 with a bar graph (now Sup. 17) showing the statistical significance of the data shown in Figure 6i.

Minor points:

"This is supported by the rapid suppression in meristem zone of coordinated motility in response to low- and high-concentration IAA treatments (Sup. Fig. 10a)."
Here I think you wanted to say the rapid augmentation/increase

Text was revised accordingly

"Primers missing TAA1 Primers describing TAA1 cloning (YUC1-2A-TAA1 cassette) are described in the methods." No. in the M&M, you cite ref 59 and 60 but these 2 refs don't use YUC1 and TAA1. They don't provide the primers you used. Plus, even if they did, it doesn't cost you anything to put them in the suppl table 2 as the other primers. It helps the reader that want to use your constructs one day. In many animal journals you would have to even add the full sequences as well as the full annotated plasmids. So please add these primers.

We added YUC1 and TAA1 cloning primers sequences. We apologies for the confusion. We thought the request was for the qPCR amplification primers.

Reviewer #2 (Remarks to the Author):

The authors have comprehensively and adequately responded to my comments, from my side the manuscript is ready to be published.

Thank you.

REVIEWERS' COMMENTS

Reviewer #1 (Remarks to the Author):

Dear Authors,

Thanks for the edits, new figures and constructive precisions added to your article.

All the best for 2021 and in your research !